# Classic GNNs are Strong Baselines: Reassessing GNNs for Node Classification

**Yuankai Luo**
Beihang University
The Hong Kong Polytechnic University
`luoyk@buaa.edu.cn`

**Lei Shi**
Beihang University
`leishi@buaa.edu.cn`

**Xiao-Ming Wu**
The Hong Kong Polytechnic University
`xiao-ming.wu@polyu.edu.hk`

## Abstract

Graph Transformers (GTs) have recently emerged as popular alternatives to traditional message-passing Graph Neural Networks (GNNs), due to their theoretically superior expressiveness and impressive performance reported on standard node classification benchmarks, often significantly outperforming GNNs. In this paper, we conduct a thorough empirical analysis to reevaluate the performance of three classic GNN models (GCN, GAT, and GraphSAGE) against GTs. Our findings suggest that the previously reported superiority of GTs may have been overstated due to suboptimal hyperparameter configurations in GNNs. Remarkably, with slight hyperparameter tuning, these classic GNN models achieve state-of-the-art performance, matching or even exceeding that of recent GTs across 17 out of the 18 diverse datasets examined. Additionally, we conduct detailed ablation studies to investigate the influence of various GNN configurations—such as normalization, dropout, residual connections, and network depth—on node classification performance. Our study aims to promote a higher standard of empirical rigor in the field of graph machine learning, encouraging more accurate comparisons and evaluations of model capabilities. Our implementation is available at https://github.com/LUOyk1999/tunedGNN.

## 1 Introduction

Node classification is a fundamental task in graph machine learning [92, 79, 54, 78, 71, 77], with high-impact applications across many fields such as social network analysis, bioinformatics, and recommendation systems. Graph Neural Networks (GNNs) [20, 28, 69, 80, 52, 33, 8, 84, 18, 9, 55, 4, 81, 60, 10, 56, 57, 70, 85, 37] have emerged as a powerful class of models for tackling the node classification task. GNNs operate by iteratively aggregating information from a node's neighbors, a process known as message passing [19], leveraging both the graph structure and node features to learn useful node representations for classification. While GNNs have achieved notable success, studies have identified several limitations, including over-smoothing [35], over-squashing [1], lack of sensitivity to heterophily [90], and challenges in capturing long-range dependencies [11].

Recently, Graph Transformers (GTs) [53, 51, 23] have gained prominence as popular alternatives to GNNs. Unlike GNNs, which primarily aggregate local neighborhood information, the Transformer architecture [68] can capture interactions between any pair of nodes via a self-attention layer. GTs have achieved significant success in graph-level tasks, e.g., graph classification involving small-scale graphs like molecular graphs [13, 82, 30, 45, 59, 6]. This success has inspired efforts [12, 17, 76,

38th Conference on Neural Information Processing Systems (NeurIPS 2024) Track on Datasets and Benchmarks.

75, 74, 88, 91, 15, 64, 7, 29, 41, 38] to utilize GTs to tackle node classification tasks, especially on large-scale graphs, addressing the aforementioned limitations of GNNs. While recent advancements in state-of-the-art GTs [12, 76] have shown promising results, it's observed that many of these models, whether explicitly or implicitly, still rely on GNNs for learning local node representations, integrating them alongside the global attention mechanisms for a more comprehensive representation.

This prompts us to reconsider: *Could the potential of message-passing GNNs for node classification have been previously underestimated?* While prior research has addressed this issue to some extent [24, 14, 73, 47, 58], these studies have limitations in terms of scope and comprehensiveness, including a restricted number and diversity of datasets, as well as an incomplete examination of hyperparameters. In this study, we comprehensively reassess the performance of GNNs for node classification, utilizing three classic GNN models—GCN [28], GAT [68], and GraphSAGE [20]—across 18 real-world benchmark datasets that include homophilous, heterophilous, and large-scale graphs. We examine the influence of key hyperparameters on GNN training, including normalization [2, 26], dropout [67], residual connections [21], and network depth. We summarize the key findings in our empirical study as follows:

- With proper hyperparameter tuning, classic GNNs can achieve highly competitive performance in node classification across homophilous and heterophilous graphs with up to millions of nodes. Notably, classic GNNs outperform state-of-the-art GTs, achieving the top rank on 17 out of 18 datasets. This indicates that the previously claimed superiority of GTs over GNNs may have been overstated, possibly due to suboptimal hyperparameter configurations in GNN evaluations.

- Our ablation studies have yielded valuable insights into GNN hyperparameters for node classification. We demonstrate that (1) normalization is essential for large-scale graphs; (2) dropout consistently proves beneficial; (3) residual connections can significantly enhance performance, especially on heterophilous graphs; and (4) GNNs on heterophilous graphs tend to perform better with deeper layers.

## 2 Classic GNNs for Node Classification

Define a graph as $\mathcal{G} = (\mathcal{V}, \mathcal{E}, \boldsymbol{X}, \boldsymbol{Y})$, where $\mathcal{V}$ denotes the set of nodes, $\mathcal{E} \subseteq \mathcal{V} \times \mathcal{V}$ represents the set of edges, $\boldsymbol{X} \in \mathbb{R}^{|\mathcal{V}| \times d}$ is the node feature matrix, with $|\mathcal{V}|$ representing the number of nodes and $d$ the dimension of the node features, and $\boldsymbol{Y} \in \mathbb{R}^{|\mathcal{V}| \times C}$ is the one-hot encoded label matrix, with $C$ being the number of classes. Let $\boldsymbol{A} \in \mathbb{R}^{|\mathcal{V}| \times |\mathcal{V}|}$ denote the adjacency matrix of $\mathcal{G}$.

**Message Passing Graph Neural Networks (GNNs)** [19] compute node representations $\boldsymbol{h}_v^l$ at each layer $l$ as:

$$\boldsymbol{h}_v^l = \text{UPDATE}^l \left( \boldsymbol{h}_v^{l-1}, \text{AGG}^l \left( \left\{ \boldsymbol{h}_u^{l-1} \mid u \in \mathcal{N}(v) \right\} \right) \right), \tag{1}$$

where $\mathcal{N}(v)$ represents the neighboring nodes adjacent to $v$, $\text{AGG}^l$ serves as the message aggregation function, and $\text{UPDATE}^l$ is the update function. Initially, each node $v$ begins with a feature vector $\boldsymbol{h}_v^0 = \boldsymbol{x}_v \in \mathbb{R}^d$. The function $\text{AGG}^l$ aggregates information from the neighbors of $v$ to update its representation. The output of the last layer $L$, i.e., $\text{GNN}(v, \boldsymbol{A}, \boldsymbol{X}) = \boldsymbol{h}_v^L$, is the representation of $v$ produced by the GNN. In this work, we focus on three classic GNNs: GCN [28], GraphSAGE [20], and GAT [68], which differ in their approach to learning the node representation $\boldsymbol{h}_v^l$.

**Graph Convolutional Networks (GCN)** [28], the standard GCN model, is formualated as:

$$\boldsymbol{h}_v^l = \sigma \left( \sum_{u \in \mathcal{N}(v) \cup \{v\}} \frac{1}{\sqrt{\hat{d}_u \hat{d}_v}} \boldsymbol{h}_u^{l-1} \boldsymbol{W}^l \right), \tag{2}$$

where $\hat{d}_v = 1 + \sum_{u \in \mathcal{N}(v)} 1$, $\sum_{u \in \mathcal{N}(v)} 1$ denotes the degree of node $v$, $\boldsymbol{W}^l$ is the trainable weight matrix in layer $l$, and $\sigma$ is the activation function, e.g., $\text{ReLU}(\cdot) = \max(0, \cdot)$.

**GraphSAGE** [20] learns node representations through a different approach:

$$\boldsymbol{h}_v^l = \sigma(\boldsymbol{h}_v^{l-1} \boldsymbol{W}_1^l + (\text{mean}_{u \in \mathcal{N}(v)} \boldsymbol{h}_u^{l-1}) \boldsymbol{W}_2^l), \tag{3}$$

where $\boldsymbol{W}_1^l$ and $\boldsymbol{W}_2^l$ are trainable weight matrices, and $\text{mean}_{u \in \mathcal{N}(v)} \boldsymbol{h}_u^{l-1}$ computes the average embedding of the neighboring nodes of $v$.

**Graph Attention Networks (GAT)** [68] employ masked self-attention to assign weights to different neighboring nodes. For an edge $(v, u) \in \mathcal{E}$, the propagation rule of GAT is defined as:

$$\alpha_{vu}^l = \frac{\exp\left(\text{LeakyReLU}\left(\mathbf{a}_l^\top \left[\boldsymbol{W}^l \boldsymbol{h}_v^{l-1} \,\|\, \boldsymbol{W}^l \boldsymbol{h}_u^{l-1}\right]\right)\right)}{\sum_{r \in \mathcal{N}(v)} \exp\left(\text{LeakyReLU}\left(\mathbf{a}_l^\top \left[\boldsymbol{W}^l \boldsymbol{h}_v^{l-1} \,\|\, \boldsymbol{W}^l \boldsymbol{h}_r^{l-1}\right]\right)\right)},$$

$$\boldsymbol{h}_v^l = \sigma\left(\sum_{u \in \mathcal{N}(v)} \alpha_{vu}^l \boldsymbol{h}_u^{l-1} \boldsymbol{W}^l\right), \tag{4}$$

where $\mathbf{a}_l$ is a trainable weight vector, $\boldsymbol{W}^l$ is a trainable weight matrix, and $\|$ represents the concatenation operation.

**Node Classification** aims to predict the labels of the unlabeled nodes. Typically, for any node $v$, the node representation generated by the last GNN layer is passed through a prediction head $g(\cdot)$, to obtain the predicted label $\hat{\boldsymbol{y}}_v = g(\text{GNN}(v, \boldsymbol{A}, \boldsymbol{X}))$. The training objective is to minimize the total loss $L(\boldsymbol{\theta}) = \sum_{v \in \mathcal{V}_{\text{train}}} \ell(\hat{\boldsymbol{y}}_v, \boldsymbol{y}_v)$ w.r.t. all nodes in the training set $\mathcal{V}_{\text{train}}$, where $\boldsymbol{y}_v$ indicates the ground-truth label of $v$ and $\boldsymbol{\theta}$ indicates the trainable GNN parameters.

**Homophilous and Heterophilous Graphs.** Node classification can be performed on both homophilous and heterophilous graphs. Homophilous graphs are characterized by edges that tend to connect nodes of the same class, while in heterophilous graphs, connected nodes may belong to different classes [58]. GNN models implicitly assume homophily in graphs [48], and it is commonly believed that due to this homophily assumption, GNNs cannot generalize well to heterophilous graphs [90, 9]. However, recent works [46, 40, 58, 42] have empirically shown that standard GCNs also work well on heterophilous graphs. In this study, we provide a comprehensive evaluation of classic GNNs for node classification on both homophilous and heterophilous graphs.

## 3 Key Hyperparameters for Training GNNs

In this section, we present an overview of the key hyperparameters for training GNNs, including normalization, dropout, residual connections, and network depth. These hyperparameters are widely utilized across different types of neural networks to improve model performance.

**Normalization.** Specifically, Layer Normalization (LN) [2] or Batch Normalization (BN) [26] can be used in every layer before the activation function $\sigma(\cdot)$. Taking GCN as an example:

$$\boldsymbol{h}_v^l = \sigma(\text{Norm}(\sum_{u \in \mathcal{N}(v) \cup \{v\}} \frac{1}{\sqrt{\hat{d}_u \hat{d}_v}} \boldsymbol{h}_u^{l-1} \boldsymbol{W}^l)). \tag{5}$$

The normalization techniques are essential for stabilizing the training process by reducing the *covariate shift*, which occurs when the distribution of each layer's node embeddings changes during training. Normalizing the node embeddings helps to maintain a more consistent distribution, allowing the use of higher learning rates and leading to faster convergence [5].

**Dropout** [67], a technique widely used in convolutional neural networks (CNNs) to address overfitting by reducing co-adaptation among hidden neurons [22, 83], has also been found to be effective in addressing similar issues in GNNs [68, 65], where the co-adaptation effects propagate and accumulate through message passing among different nodes. Typically, dropout is applied to the feature embeddings after the activation function:

$$\boldsymbol{h}_v^l = \text{Dropout}(\sigma(\text{Norm}(\sum_{u \in \mathcal{N}(v) \cup \{v\}} \frac{1}{\sqrt{\hat{d}_u \hat{d}_v}} \boldsymbol{h}_u^{l-1} \boldsymbol{W}^l))). \tag{6}$$

**Residual Connections** [21] significantly enhance CNN performance by connecting layer inputs directly to outputs, thereby alleviating the vanishing gradient issue. They were first adopted by the seminal GCN paper [28] and subsequently incorporated into DeepGCNs [33] to boost performance. Formally, linear residual connections can be integrated into GNNs as follows:

$$\boldsymbol{h}_v^l = \text{Dropout}(\sigma(\text{Norm}(\boldsymbol{h}_v^{l-1} \boldsymbol{W_r}^l + \sum_{u \in \mathcal{N}(v) \cup \{v\}} \frac{1}{\sqrt{\hat{d}_u \hat{d}_v}} \boldsymbol{h}_u^{l-1} \boldsymbol{W}^l))), \tag{7}$$

Table 1: Overview of the datasets used for node classification.

| Dataset | Type | # nodes | # edges | # Features | Classes | Metric |
|---------|------|---------|---------|-----------|---------|--------|
| Cora | Homophily | 2,708 | 5,278 | 1,433 | 7 | Accuracy |
| CiteSeer | Homophily | 3,327 | 4,522 | 3,703 | 6 | Accuracy |
| PubMed | Homophily | 19,717 | 44,324 | 500 | 3 | Accuracy |
| Computer | Homophily | 13,752 | 245,861 | 767 | 10 | Accuracy |
| Photo | Homophily | 7,650 | 119,081 | 745 | 8 | Accuracy |
| CS | Homophily | 18,333 | 81,894 | 6,805 | 15 | Accuracy |
| Physics | Homophily | 34,493 | 247,962 | 8,415 | 5 | Accuracy |
| WikiCS | Homophily | 11,701 | 216,123 | 300 | 10 | Accuracy |
| Squirrel | Heterophily | 2,223 | 46,998 | 2,089 | 5 | Accuracy |
| Chameleon | Heterophily | 890 | 8,854 | 2,325 | 5 | Accuracy |
| Roman-Empire | Heterophily | 22,662 | 32,927 | 300 | 18 | Accuracy |
| Amazon-Ratings | Heterophily | 24,492 | 93,050 | 300 | 5 | Accuracy |
| Minesweeper | Heterophily | 10,000 | 39,402 | 7 | 2 | ROC-AUC |
| Questions | Heterophily | 48,921 | 153,540 | 301 | 2 | ROC-AUC |
| ogbn-proteins | Homophily (Large graphs) | 132,534 | 39,561,252 | 8 | 2 | ROC-AUC |
| ogbn-arxiv | Homophily (Large graphs) | 169,343 | 1,166,243 | 128 | 40 | Accuracy |
| ogbn-products | Homophily (Large graphs) | 2,449,029 | 61,859,140 | 100 | 47 | Accuracy |
| pokec | Heterophily (Large graphs) | 1,632,803 | 30,622,564 | 65 | 2 | Accuracy |

where $W_r^l$ is a trainable weight matrix. This configuration mitigates gradient instabilities and enhances GNN expressiveness [80], addressing the over-smoothing [35] and oversquashing [1] issues since the linear component $(h_u^{l-1} W_r{}^l)$ helps to preserve distinguishable node representations [73].

**Network Depth.** Deeper network architectures, such as deep CNNs [21, 25], are capable of extracting more complex, high-level features from data, potentially leading to better performance on various prediction tasks. However, GNNs face unique challenges with depth, such as over-smoothing [35], where node representations become indistinguishable with increased network depth. Consequently, in practice, most GNNs adopt a shallow architecture, typically consisting of 2 to 5 layers. While previous research, such as DeepGCN [33] and DeeperGCN [34], advocates the use of deep GNNs with up to 56 and 112 layers, our findings indicate that comparable performance can be achieved with significantly shallower GNN architectures, typically ranging from 2 to 10 layers.

## 4 Experimental Setup for Node Classification

**Datasets.** Table 1 presents a summary of the statistics and characteristics of the datasets.

- **Homophilous Graphs. Cora**, **CiteSeer**, and **PubMed** are three widely used citation networks [62]. We follow the semi-supervised setting of [28] for data splits and metrics. Additionally, **Computer** and **Photo** [63] are co-purchase networks where nodes represent goods and edges indicate that the connected goods are frequently bought together. **CS** and **Physics** [63] are co-authorship networks where nodes denote authors and edges represent that the authors have co-authored at least one paper. We adhere to the widely accepted practice of training/validation/test splits of 60%/20%/20% and metric of accuracy [7, 64, 12]. Furthermore, we utilize the **WikiCS** dataset and use the official splits and metrics provided in [50].
- **Heterophilous Graphs. Squirrel** and **Chameleon** [61] are two well-known page-page networks that focus on specific topics in Wikipedia. According to the heterophilous graphs benchmarking paper [58], the original split of these datasets introduces overlapping nodes between training and testing, leading to the proposal of a new data split that filters out the overlapping nodes. We use its provided split and its metrics for evaluation. Additionally, we utilize four other heterophilous datasets proposed by the same source [58]: **Roman-Empire**, where nodes correspond to words in the Roman Empire Wikipedia article and edges connect sequential or syntactically linked words; **Amazon-Ratings**, where nodes represent products and edges connect frequently co-purchased items; **Minesweeper**, a synthetic dataset where nodes are cells in a $100 \times 100$ grid and edges connect neighboring cells; and **Questions**, where nodes represent users from the Yandex Q question-answering website and edges connect users who interacted through answers. All splits and evaluation metrics are consistent with those proposed in the source.

Table 2: Node classification results over homophilous graphs (%). * indicates our implementation, while other results are taken from [12, 76]. The top 1st, 2nd and 3rd results are highlighted.

| | Cora | CiteSeer | PubMed | Computer | Photo | CS | Physics | WikiCS |
|---|---|---|---|---|---|---|---|---|
| # nodes | 2,708 | 3,327 | 19,717 | 13,752 | 7,650 | 18,333 | 34,493 | 11,701 |
| # edges | 5,278 | 4,732 | 44,324 | 245,861 | 119,081 | 81,894 | 247,962 | 216,123 |
| Metric | Accuracy↑ | Accuracy↑ | Accuracy↑ | Accuracy↑ | Accuracy↑ | Accuracy↑ | Accuracy↑ | Accuracy↑ |
| GraphGPS | 82.84 ±1.03 | 72.73 ±1.23 | 79.94 ±0.26 | 91.19 ±0.54 | 95.06 ±0.13 | 93.93 ±0.12 | 97.12 ±0.19 | 78.66 ±0.49 |
| GraphGPS* | 83.87 ±0.96 | 72.73 ±1.23 | 79.94 ±0.26 | 91.79 ±0.63 | 94.89 ±0.14 | 94.04 ±0.21 | 96.71 ±0.15 | 78.66 ±0.49 |
| NAGphormer | 82.12 ±1.18 | 71.47 ±1.30 | 79.73 ±0.28 | 91.22 ±0.14 | 95.49 ±0.11 | 95.75 ±0.09 | 97.34 ±0.03 | 77.16 ±0.72 |
| NAGphormer* | 80.92 ±1.17 | 70.59 ±0.89 | 80.14 ±1.06 | 91.69 ±0.30 | 96.14 ±0.16 | 95.85 ±0.16 | 97.35 ±0.12 | 77.92 ±0.93 |
| Exphormer | 82.77 ±1.38 | 71.63 ±1.19 | 79.46 ±0.35 | 91.47 ±0.17 | 95.35 ±0.22 | 94.93 ±0.01 | 96.89 ±0.09 | 78.54 ±0.49 |
| Exphormer* | 83.29 ±1.36 | 71.85 ±1.11 | 79.67 ±0.73 | 91.80 ±0.35 | 95.69 ±0.39 | 95.92 ±0.25 | 97.06 ±0.13 | 79.38 ±0.62 |
| GOAT | 83.18 ±1.27 | 71.99 ±1.26 | 79.13 ±0.38 | 90.96 ±0.90 | 92.96 ±1.48 | 94.21 ±0.38 | 96.24 ±0.24 | 77.00 ±0.77 |
| GOAT* | 83.26 ±1.24 | 72.21 ±1.29 | 80.06 ±0.67 | 92.29 ±0.37 | 94.33 ±0.21 | 93.81 ±0.19 | 96.47 ±0.16 | 77.96 ±0.63 |
| NodeFormer | 82.20 ±0.90 | 72.50 ±1.10 | 79.90 ±1.00 | 86.98 ±0.62 | 93.46 ±0.35 | 95.64 ±0.22 | 96.45 ±0.28 | 74.73 ±0.94 |
| NodeFormer* | 82.73 ±0.75 | 72.37 ±1.20 | 79.59 ±0.92 | 87.29 ±0.58 | 93.43 ±0.56 | 95.69 ±0.27 | 96.48 ±0.34 | 75.13 ±0.93 |
| SGFormer | 84.50 ±0.80 | 72.60 ±0.20 | 80.30 ±0.60 | 91.99 ±0.76 | 95.10 ±0.47 | 94.78 ±0.20 | 96.60 ±0.18 | 73.46 ±0.56 |
| SGFormer* | 84.82 ±0.85 | 72.72 ±1.15 | 80.60 ±0.49 | 92.42 ±0.66 | 95.58 ±0.36 | 95.71 ±0.24 | 96.75 ±0.26 | 80.05 ±0.46 |
| Polynormer | 83.25 ±0.93 | 72.31 ±0.78 | 79.24 ±0.43 | 93.68 ±0.21 | 96.46 ±0.26 | 95.53 ±0.16 | 97.27 ±0.08 | 80.10 ±0.67 |
| Polynormer* | 83.43 ±0.89 | 72.19 ±0.83 | 79.35 ±0.73 | 93.78 ±0.10 | 96.57 ±0.23 | 95.42 ±0.19 | 97.18 ±0.11 | 80.26 ±0.92 |
| GCN | 81.60 ±0.40 | 71.60 ±0.40 | 78.80 ±0.60 | 89.65 ±0.52 | 92.70 ±0.20 | 92.92 ±0.12 | 96.18 ±0.07 | 77.47 ±0.85 |
| GCN* | 85.10 ±0.67 3.50↑ | 73.14 ±0.67 1.54↑ | 81.12 ±0.52 2.32↑ | 93.99 ±0.12 4.34↑ | 96.10 ±0.46 3.40↑ | 96.17 ±0.06 3.25↑ | 97.46 ±0.10 1.28↑ | 80.30 ±0.62 2.83↑ |
| GraphSAGE | 82.68 ±0.47 | 71.93 ±0.85 | 79.41 ±0.53 | 91.20 ±0.29 | 94.59 ±0.14 | 93.91 ±0.13 | 96.49 ±0.06 | 74.77 ±0.95 |
| GraphSAGE* | 83.88 ±0.65 1.20↑ | 72.26 ±0.55 0.33↑ | 79.72 ±0.50 0.31↑ | 93.25 ±0.14 2.05↑ | 96.78 ±0.23 2.19↑ | 96.38 ±0.11 2.47↑ | 97.19 ±0.05 0.70↑ | 80.69 ±0.31 5.92↑ |
| GAT | 83.00 ±0.70 | 72.10 ±1.10 | 79.00 ±0.40 | 90.78 ±0.13 | 93.87 ±0.11 | 93.61 ±0.14 | 96.17 ±0.08 | 76.91 ±0.82 |
| GAT* | 84.46 ±0.55 1.46↑ | 72.22 ±0.84 0.12↑ | 80.28 ±0.64 1.28↑ | 94.09 ±0.37 3.31↑ | 96.60 ±0.33 2.73↑ | 96.21 ±0.14 2.60↑ | 97.25 ±0.06 1.08↑ | 81.07 ±0.54 4.16↑ |

- **Large-scale Graphs.** We consider a collection of large graphs released recently by the Open Graph Benchmark (OGB) [24]: **ogbn-arxiv**, **ogbn-proteins**, and **ogbn-products**, with node numbers ranging from 0.16M to 2.4M. We maintain all the OGB standard evaluation settings. Additionally, we analyze performance on the social network **pokec** [32], which has 1.6M nodes, following the evaluation settings of [12].

**Baselines.** Our main focus lies on classic GNNs: **GCN** [28], **GraphSAGE** [20], **GAT** [68], the state-of-the-art scalable GTs: **SGFormer** [76], **Polynormer** [12], **GOAT** [29], **NodeFormer** [75], **NAGphormer** [7], and powerful GTs: **GraphGPS** [59] and **Exphormer** [64]. Furthermore, various other GTs like [17, 15, 38, 86, 31, 3, 6, 82, 13] exist in related surveys [23, 53], empirically shown to be inferior to the GTs we compared against for node classification tasks. For heterophilous graphs, We also consider five models designed for node classification under heterophily following [58]: **H2GCN** [90], **CPGNN** [89], **GPRGNN** [9], **FSGNN** [49], **GloGNN** [36]. Note that we adopt the empirically optimal Polynormer variant (Polynormer-r), which demonstrates superior performance over advanced GNNs such as LINKX [37] and OrderedGNN [66]. We report the performance results of baselines primarily from [12, 76, 58], with the remaining obtained from their respective original papers or official leaderboards whenever possible, as those results are obtained by well-tuned models.

**Hyperparameter Configurations.** We conduct hyperparameter tuning on classic GNNs, consistent with the hyperparameter search space of Polynormer [12]. Specifically, we utilize the Adam optimizer [27] with a learning rate from $\{0.001, 0.005, 0.01\}$ and an epoch limit of 2500. And we tune the hidden dimension from $\{64, 256, 512\}$. As discussed in Section 3, we focus on whether to use normalization (BN or LN), residual connections, and dropout rates from $\{0.2, 0.3, 0.5, 0.7\}$, the number of layers from $\{1, 2, 3, 4, 5, 6, 7, 8, 9, 10\}$. Additionally, we retrain all baseline GTs using the same hyperparameter search space and training environments as the classic GNNs. For hyperparameters specific to each GT, which are not present in the classic GNNs, we tune them according to the search space specified in the original GT paper. We report mean scores and standard deviations after 5 independent runs with different initializations. **Model**\* denotes our implementation.

Detailed experimental setup and hyperparameters are provided in Appendix A.

## 5 Empirical Findings

### 5.1 Performance of Classic GNNs in Node Classification

In this subsection, we provide a detailed analysis of the performance of the three classic GNNs compared to state-of-the-art GTs in node classification tasks. Our experimental results across homophilous (Table 2), heterophilous (Table 3), and large-scale graphs (Table 4) reveal that classic GNNs often outperform or match the performance of advanced GTs across 18 datasets. Notably,

Table 3: Node classification results on heterophilous graphs (%). * indicates our implementation, while other results are taken from [12, 76, 58]. The top $1^{st}$, $2^{nd}$ and $3^{rd}$ results are highlighted.

| | Squirrel | Chameleon | Amazon-Ratings | Roman-Empire | Minesweeper | Questions |
|---|---|---|---|---|---|---|
| # nodes | 2223 | 890 | 24,492 | 22,662 | 10,000 | 48,921 |
| # edges | 46,998 | 8,854 | 93,050 | 32,927 | 39,402 | 153,540 |
| Metric | Accuracy↑ | Accuracy↑ | Accuracy↑ | Accuracy↑ | ROC-AUC↑ | ROC-AUC↑ |
| H2GCN | 35.10 ±1.15 | 26.75 ±3.64 | 36.47 ±0.23 | 60.11 ±0.52 | 89.71 ±0.31 | 63.59 ±1.46 |
| CPGNN | 30.04 ±2.03 | 33.00 ±3.15 | 39.79 ±0.77 | 63.96 ±0.62 | 52.03 ±5.46 | 65.96 ±1.95 |
| GPRGNN | 38.95 ±1.99 | 39.93 ±3.30 | 44.88 ±0.27 | 64.85 ±0.27 | 86.24 ±0.61 | 55.48 ±0.91 |
| FSGNN | 35.92 ±1.32 | 40.61 ±2.97 | 52.74 ±0.83 | 79.92 ±0.56 | 90.08 ±0.70 | 78.86 ±0.92 |
| GloGNN | 35.11 ±1.24 | 25.90 ±3.58 | 36.89 ±0.14 | 59.63 ±0.69 | 51.08 ±1.23 | 65.74 ±1.19 |
| GraphGPS | 39.67 ±2.84 | 40.79 ±4.03 | 53.10 ±0.42 | 82.00 ±0.61 | 90.63 ±0.67 | 71.73 ±1.47 |
| **GraphGPS*** | 39.81 ±2.28 | 41.55 ±3.91 | 53.27 ±0.66 | 82.72 ±0.68 | 90.75 ±0.89 | 72.56 ±1.33 |
| NodeFormer | 38.52 ±1.57 | 34.73 ±4.14 | 43.86 ±0.35 | 64.49 ±0.73 | 86.71 ±0.88 | 74.27 ±1.46 |
| **NodeFormer*** | 38.89 ±2.67 | 36.38 ±3.85 | 43.79 ±0.57 | 74.83 ±0.81 | 87.71 ±0.69 | 75.02 ±1.61 |
| SGFormer | 41.80 ±2.27 | 44.93 ±3.91 | 48.01 ±0.49 | 79.10 ±0.32 | 90.89 ±0.58 | 72.15 ±1.31 |
| **SGFormer*** | 42.65 ±2.41 | 45.21 ±3.72 | 54.14 ±0.62 | 80.01 ±0.44 | 91.42 ±0.41 | 73.81 ±0.59 |
| Polynormer | 40.87 ±1.96 | 41.82 ±3.45 | 54.81 ±0.49 | 92.55 ±0.37 | 97.46 ±0.36 | 78.92 ±0.89 |
| **Polynormer*** | 41.97 ±2.14 | 41.97 ±3.18 | 54.96 ±0.22 | 92.66 ±0.60 | 97.49 ±0.48 | 78.94 ±0.78 |
| GCN | 38.67 ±1.84 | 41.31 ±3.05 | 48.70 ±0.63 | 73.69 ±0.74 | 89.75 ±0.52 | 76.09 ±1.27 |
| **GCN*** | 45.01 ±1.63 **6.34↑** | 46.29 ±3.40 **4.98↑** | 53.80 ±0.60 **5.10↑** | 91.27 ±0.20 **17.58↑** | 97.86 ±0.24 **8.11↑** | 79.02 ±0.60 **2.93↑** |
| GraphSAGE | 36.09 ±1.99 | 37.77 ±4.14 | 53.63 ±0.39 | 85.74 ±0.67 | 93.51 ±0.57 | 76.44 ±0.62 |
| **GraphSAGE*** | 40.78 ±1.47 **4.69↑** | 44.81 ±4.74 **7.04↑** | 55.40 ±0.21 **1.77↑** | 91.06 ±0.27 **5.32↑** | 97.77 ±0.62 **4.26↑** | 77.21 ±1.28 **0.77↑** |
| GAT | 35.62 ±2.06 | 39.21 ±3.08 | 52.70 ±0.62 | 88.75 ±0.41 | 93.91 ±0.35 | 76.79 ±0.71 |
| **GAT*** | 41.73 ±2.07 **6.11↑** | 44.13 ±4.17 **4.92↑** | 55.54 ±0.51 **2.84↑** | 90.63 ±0.14 **1.88↑** | 97.73 ±0.73 **3.82↑** | 77.95 ±0.51 **1.16↑** |

among the 18 datasets evaluated, classic GNNs achieve the top rank on 17 of them, showcasing their robust competitiveness. We highlight our main observations below.

> **Observations on Homophilous Graphs (Table 2).** Classic GNNs, with only slight adjustments to hyperparameters, are highly competitive in node classification tasks on homophilous graphs, often outperforming state-of-the-art graph transformers in many cases.

While previously reported results show that most advanced GTs outperform classic GNN on homophilous graphs [12, 76], our implementation of classic GNNs can place within the top two for four datasets, with GCN* and GAT* demonstrating near-consistent top performances. Specifically, on CS and WikiCS, classic GNNs experience about a 3% accuracy increase, achieving top-three performances. On WikiCS, the accuracy of GAT* increases by 4.16%, moving it from seventh to first place, surpassing the leading GT, Polynormer. Similarly, on Photo and CS, GraphSAGE* outperforms Polynormer and SGFormer, establishing itself as the top model. On Cora, CiteSeer, PubMed, and Physics, tuning yields significant performance improvements for GCN*, with accuracy increases ranging from 1.54% to 3.50%, positioning GCN* as the highest-performing model despite its initial lower accuracy compared to advanced GTs.

> **Observations on Heterophilous Graphs (Table 3).** Our implementation has significantly enhanced the previously reported best results of classic GNNs on heterophilous graphs, surpassing specialized GNN models tailored for such graphs and even outperforming the leading graph transformer architectures. This advancement not only supports but also strengthens the findings in [58] that conventional GNNs are strong contenders for heterophilous graphs, challenging the prevailing assumption that they are primarily suited for homophilous graph structures.

The three classic GNNs secure top positions on five out of six heterophilous graphs. Specifically, on well-known page-page networks like Chameleon and Squirrel, our implementation enhances the accuracy of GCN* by 4.98% and 6.34% respectively, elevating it to the first place among all models. Similarly, on larger heterophilous graphs such as Minesweeper and Questions, GCN* also exhibits the highest performance, highlighting the superiority of its local message-passing mechanism over GTs' global attention. On Roman-Empire, a 17.58% increase is observed in the performance of GCN*. Interestingly, we find that improvements primarily stem from residual connections, which are further analyzed in our ablation study (see Section 5.2).

Table 4: Node classification results on large-scale graphs (%). * indicates our implementation, while other results are taken from [12, 76]. The top **1st**, **2nd** and **3rd** results are highlighted. OOM means out of memory.

| | ogbn-proteins | ogbn-arxiv | ogbn-products | pokec |
|---|---|---|---|---|
| # nodes | 132,534 | 169,343 | 2,449,029 | 1,632,803 |
| # edges | 39,561,252 | 1,166,243 | 61,859,140 | 30,622,564 |
| Metric | ROC-AUC↑ | Accuracy↑ | Accuracy↑ | Accuracy↑ |
| GraphGPS | 76.83 $\pm$ 0.26 | 70.97 $\pm$ 0.41 | OOM | OOM |
| **GraphGPS*** | 77.15 $\pm$ 0.64 | 71.23 $\pm$ 0.59 | OOM | OOM |
| NAGphormer | 73.61 $\pm$ 0.33 | 70.13 $\pm$ 0.55 | 73.55 $\pm$ 0.21 | 76.59 $\pm$ 0.25 |
| **NAGphormer*** | 72.17 $\pm$ 0.45 | 70.88 $\pm$ 0.24 | 74.63 $\pm$ 0.29 | 75.92 $\pm$ 0.68 |
| Exphormer | 74.58 $\pm$ 0.26 | 72.44 $\pm$ 0.28 | OOM | OOM |
| **Exphormer*** | 77.62 $\pm$ 0.33 | 72.32 $\pm$ 0.36 | OOM | OOM |
| GOAT | 74.18 $\pm$ 0.37 | 72.41 $\pm$ 0.40 | 82.00 $\pm$ 0.43 | 66.37 $\pm$ 0.94 |
| **GOAT*** | 79.31 $\pm$ 0.42 | 72.76 $\pm$ 0.29 | 82.27 $\pm$ 0.56 | 72.64 $\pm$ 0.67 |
| NodeFormer | 77.45 $\pm$ 1.15 | 59.90 $\pm$ 0.42 | 72.93 $\pm$ 0.13 | 71.00 $\pm$ 1.30 |
| **NodeFormer*** | 77.86 $\pm$ 0.84 | 67.78 $\pm$ 0.28 | 73.96 $\pm$ 0.30 | 71.00 $\pm$ 1.30 |
| SGFormer | 79.53 $\pm$ 0.38 | 72.63 $\pm$ 0.13 | 74.16 $\pm$ 0.31 | 73.76 $\pm$ 0.24 |
| **SGFormer*** | **79.92** $\pm$ 0.48 | 72.76 $\pm$ 0.33 | 81.54 $\pm$ 0.43 | 82.44 $\pm$ 0.76 |
| Polynormer | 78.97 $\pm$ 0.47 | 73.46 $\pm$ 0.16 | 83.82 $\pm$ 0.11 | 86.10 $\pm$ 0.05 |
| **Polynormer*** | 79.53 $\pm$ 0.67 | **73.40** $\pm$ 0.22 | **83.82** $\pm$ 0.11 | **86.06** $\pm$ 0.25 |
| GCN | 72.51 $\pm$ 0.35 | 71.74 $\pm$ 0.29 | 75.64 $\pm$ 0.21 | 75.45 $\pm$ 0.17 |
| **GCN*** | 77.29 $\pm$ 0.46 **4.78**↑ | **73.53** $\pm$ 0.12 **1.79**↑ | **82.33** $\pm$ 0.19 **6.69**↑ | **86.33** $\pm$ 0.17 **10.88**↑ |
| GraphSAGE | 77.68 $\pm$ 0.20 | 71.49 $\pm$ 0.27 | 78.29 $\pm$ 0.16 | 75.63 $\pm$ 0.38 |
| **GraphSAGE*** | **82.21** $\pm$ 0.32 **4.53**↑ | 73.00 $\pm$ 0.28 **1.51**↑ | **83.89** $\pm$ 0.36 **5.60**↑ | 85.97 $\pm$ 0.21 **10.34**↑ |
| GAT | 72.02 $\pm$ 0.44 | 71.95 $\pm$ 0.36 | 79.45 $\pm$ 0.59 | 72.23 $\pm$ 0.18 |
| **GAT*** | **85.01** $\pm$ 0.46 **12.99**↑ | **73.30** $\pm$ 0.18 **1.35**↑ | 80.99 $\pm$ 0.16 **1.54**↑ | **86.19** $\pm$ 0.23 **13.96**↑ |

> **Observations on Large-scale Graphs (Table 4).** Our implementation has significantly enhanced the previously reported results of classic GNNs, with some cases showing double-digit increases in accuracy. It has achieved the best results across these large graph datasets, either homophilous or heterophilous, and has outperformed state-of-the-art graph transformers. This indicates that message passing remains highly effective for learning node representations on large-scale graphs.

Our implementation of classic GNNs demonstrate superior performance consistently, achieving top rankings across all four large-scale datasets included in our study. Notably, GCN* emerges as the leading model on ogbn-arxiv and pokec, surpassing all evaluated advanced GTs. Furthermore, on pokec, all three classic GNNs achieve over 10% performance increases by our implementation. For ogbn-proteins, an absolute improvement of 12.99% is observed in the performance of GAT*, significantly surpassing SGFormer by 5.09%. Similarly, on ogbn-products, GraphSAGE* demonstrates a significant performance increase, securing the best performance among all evaluated models. In summary, a basic GNN can achieve the best known results on large-scale graphs, suggesting that current GTs have not yet addressed GNN issues such as over-smoothing and long-range dependencies.

## 5.2 Influence of Hyperparameters on the Performance of GNNs

To examine the unique contributions of different hyperparameters in explaining the enhanced performance of classic GNNs, we conduct a series of ablation analysis by selectively removing elements such as normalization, dropout, residual connections, and network depth from GCN*, GraphSAGE*, and GAT*. The effect of these ablations is assessed across homophilous (see Table 5), heterphilous (see Table 6), and large-scale graphs (see Table 7). Our findings, which we detail below, indicate that the ablation of single components affects model accuracy in distinct ways.

**Observation 1: Normalization (either BN or LN) is important for node classification on large-scale graphs but less significant on smaller-scale graphs.**

Table 5: Ablation study on homophilous graphs (%). - indicates that the corresponding hyperparameter is not used in GNN*, as it empirically leads to inferior performance.

| | Cora | CiteSeer | PubMed | Computer | Photo | CS | Physics | WikiCS |
|---|---|---|---|---|---|---|---|---|
| Metric | Accuracy↑ | Accuracy↑ | Accuracy↑ | Accuracy↑ | Accuracy↑ | Accuracy↑ | Accuracy↑ | Accuracy↑ |
| **GCN*** | **85.10** ±0.67 | **73.14** ±0.67 | **81.12** ±0.52 | **93.99** ±0.12 | **96.10** ±0.46 | **96.17** ±0.06 | **97.46** ±0.10 | **80.30** ±0.62 |
| (-) Normalization | - | - | - | 92.60 ±0.14 | 95.48 ±0.36 | 95.30 ±0.05 | 97.16 ±0.11 | 79.67 ±0.52 |
| (-) Dropout | 83.46 ±1.16 | 71.40 ±0.35 | 80.14 ±0.55 | 93.78 ±0.26 | 95.31 ±0.10 | 95.95 ±0.14 | 97.30 ±0.06 | 79.84 ±0.86 |
| (-) Residual Connections | - | - | - | - | 94.43 ±0.16 | 94.71 ±0.12 | 96.56 ±0.18 | - |
| **GraphSAGE*** | **83.88** ±0.65 | **72.26** ±0.55 | **79.72** ±0.50 | **93.25** ±0.14 | **96.78** ±0.23 | **96.38** ±0.11 | **97.19** ±0.05 | **80.69** ±0.31 |
| (-) Normalization | - | - | - | 92.77 ±0.63 | 95.51 ±0.46 | 95.42 ±0.13 | 96.97 ±0.07 | 80.08 ±0.85 |
| (-) Dropout | 82.78 ±0.54 | 71.02 ±1.34 | 77.02 ±0.83 | 92.02 ±0.35 | 96.03 ±0.27 | 96.11 ±0.17 | 97.07 ±0.09 | 79.89 ±0.39 |
| (-) Residual Connections | - | - | - | - | 96.47 ±0.11 | 95.73 ±0.13 | 97.09 ±0.04 | - |
| **GAT*** | **84.46** ±0.55 | **72.22** ±0.84 | **80.28** ±0.64 | **94.09** ±0.37 | **96.60** ±0.33 | **96.21** ±0.14 | **97.25** ±0.06 | **81.07** ±0.54 |
| (-) Normalization | - | - | - | 93.22 ±1.27 | 96.09 ±0.20 | 95.13 ±0.16 | 97.08 ±0.04 | 80.34 ±0.41 |
| (-) Dropout | 83.30 ±1.34 | 71.22 ±1.02 | 78.36 ±1.22 | 93.14 ±0.29 | 96.36 ±0.26 | 96.05 ±0.09 | 97.01 ±0.05 | 79.46 ±0.32 |
| (-) Residual Connections | 82.62 ±1.08 | 71.60 ±0.89 | - | - | 95.05 ±0.14 | 94.47 ±0.09 | 96.44 ±0.03 | 80.32 ±0.90 |

Table 6: Ablation study on heterophilous graphs (%).

| | Squirrel | Chameleon | Amazon-Ratings | Roman-Empire | Minesweeper | Questions |
|---|---|---|---|---|---|---|
| Metric | Accuracy↑ | Accuracy↑ | Accuracy↑ | Accuracy↑ | ROC-AUC↑ | ROC-AUC↑ |
| **GCN*** | **45.01** ±1.63 | **46.29** ±3.40 | **53.80** ±0.60 | **91.27** ±0.20 | **97.86** ±0.24 | **79.02** ±0.60 |
| (-) Normalization | 44.13 ±2.03 | - | 53.68 ±0.82 | 90.53 ±0.33 | 96.94 ±1.96 | - |
| (-) Dropout | 42.89 ±1.28 | 45.28 ±4.78 | 51.37 ±0.34 | 85.10 ±0.61 | 94.28 ±2.29 | 76.58 ±0.40 |
| (-) Residual Connections | 43.14 ±1.82 | - | 51.14 ±0.34 | 74.84 ±0.62 | 86.45 ±0.89 | 75.87 ±4.47 |
| **GraphSAGE*** | **40.78** ±1.47 | **44.81** ±4.74 | **55.40** ±0.21 | **91.06** ±0.27 | **97.77** ±0.62 | **77.21** ±1.28 |
| (-) Normalization | 40.27 ±2.27 | 44.02 ±3.53 | 54.41 ±0.30 | 90.58 ±0.24 | 97.64 ±0.41 | 76.17 ±0.41 |
| (-) Dropout | 38.83 ±1.94 | 43.11 ±3.36 | 51.12 ±0.66 | 84.49 ±0.35 | 93.83 ±0.38 | 76.36 ±1.50 |
| (-) Residual Connections | 40.06 ±2.31 | 41.85 ±3.86 | 53.52 ±0.19 | - | 96.64 ±0.85 | - |
| **GAT*** | **41.73** ±2.07 | **44.13** ±4.17 | **55.54** ±0.51 | **90.63** ±0.14 | **97.73** ±0.73 | **77.95** ±0.51 |
| (-) Normalization | 41.08 ±1.63 | 43.25 ±3.84 | 54.85 ±0.39 | 89.69 ±0.39 | 97.42 ±0.85 | 76.32 ±0.24 |
| (-) Dropout | 39.81 ±3.15 | 41.19 ±2.36 | 51.48 ±0.28 | 82.47 ±0.70 | 92.26 ±4.63 | 76.19 ±0.88 |
| (-) Residual Connections | 38.46 ±1.96 | 42.57 ±3.66 | 51.08 ±0.49 | 85.15 ±0.82 | 92.83 ±1.61 | 75.17 ±0.71 |

We observe that the ablation of normalization does not lead to substantial deviations on small graphs. However, normalization becomes consistently crucial on large-scale graphs, where its ablation results in accuracy reductions of 4.79% and 4.69% for GraphSAGE* and GAT* respectively on ogbn-proteins. We believe this is because large graphs display a wider variety of node features, resulting in different data distributions across the graph. Normalization aids in standardizing these features during training, ensuring a more stable distribution.

**Observation 2: Dropout is consistently found to be essential for node classification.**

Our analysis highlights the crucial role of dropout in maintaining the performance of classic GNNs on both homophilous and heterophilous graphs, with its ablation contributing to notable accuracy declines—for instance, a 2.70% decrease for GraphSAGE* on PubMed and a 6.57% decrease on Roman-Empire. This trend persists in large-scale datasets, where the ablation of dropout leads to a 2.44% and 2.53% performance decline for GCN* and GAT* respectively on ogbn-proteins.

**Observation 3: Residual connections can significantly boost performance on specific datasets, exhibiting a more pronounced effect on heterophilous graphs than on homophilous graphs.**

While the ablation of residual connections on homophilous graphs does not consistently lead to a significant performance decrease, with observed differences around 2% on Cora, Photo, and CS, the impact is more substantial on large-scale graphs such as ogbn-proteins and pokec. The effect is even more dramatic on heterophilous graphs, with the classic GNNs exhibiting the most significant accuracy reduction on Roman-Empire, for instance, a 16.43% for GCN* and 5.48% for GAT*. Similarly, on Minesweeper, significant performance drops were observed, emphasizing the critical importance of residual connections, particularly on heterophilous graphs. The complex structures of these graphs often necessitate deeper layers to effectively capture the diverse relationships between nodes. In such contexts, residual connections are essential for model training.

Table 7: Ablation study on large-scale graphs (%).

|  | ogbn-proteins | ogbn-arxiv | ogbn-products | pokec |
|---|---|---|---|---|
| Metric | ROC-AUC↑ | Accuracy↑ | Accuracy↑ | Accuracy↑ |
| **GCN**[*] | **77.29** ±0.46 | **73.53** ±0.12 | **82.33** ±0.19 | **86.33** ±0.17 |
| (-) Normalization | 74.48 ±1.13 | 71.53 ±0.14 | 80.01 ±0.48 | 85.21 ±0.23 |
| (-) Dropout | 74.85 ±0.87 | 72.06 ±0.13 | 79.30 ±0.37 | 84.47 ±0.38 |
| (-) Residual Connections | 73.19 ±1.46 | 72.91 ±0.17 | - | 79.59 ±0.97 |
| **GraphSAGE**[*] | **82.21** ±0.32 | **73.00** ±0.28 | **83.89** ±0.36 | **85.97** ±0.21 |
| (-) Normalization | 77.42 ±0.98 | 71.13 ±0.27 | 82.12 ±0.31 | 84.95 ±0.33 |
| (-) Dropout | 80.52 ±0.49 | 71.30 ±0.21 | 80.36 ±0.43 | 83.06 ±0.28 |
| (-) Residual Connections | 81.75 ±0.53 | 72.22 ±0.49 | - | 85.81 ±0.45 |
| **GAT**[*] | **85.01** ±0.46 | **73.30** ±0.18 | **80.99** ±0.16 | **86.19** ±0.23 |
| (-) Normalization | 80.32 ±0.83 | 71.33 ±0.29 | 78.62 ±0.33 | 84.63 ±0.64 |
| (-) Dropout | 82.48 ±0.34 | 71.68 ±0.32 | 77.68 ±0.21 | 85.12 ±0.49 |
| (-) Residual Connections | 82.43 ±0.75 | 72.47 ±0.34 | - | 81.37 ±0.87 |

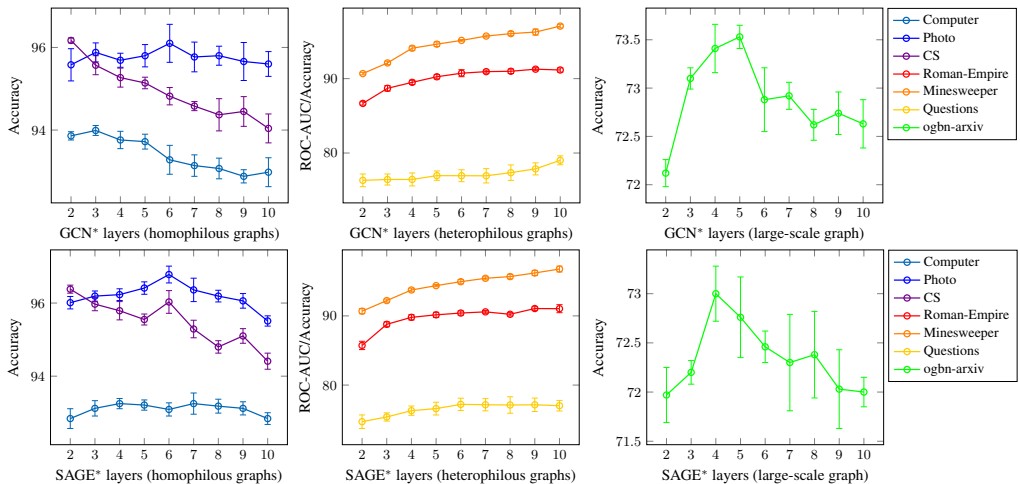

Figure 1: Ablation studies of the number of layers showing, from left to right, results for homophilous graphs, heterophilous graphs, and large-scale graphs, respectively.

**Observation 4: Deeper networks generally lead to greater performance gains on heterophilous graphs compared to homophilous graphs.**

As demonstrated in Figure 1, the performance trends for GCN[*] and GraphSAGE[*] are consistent across different graph types. On homophilous graphs and ogbn-arxiv, both models achieve optimal performance with a range of 2 to 6 layers. In contrast, on heterophilous graphs, their performance improves with an increasing number of layers, indicating that deeper networks are more beneficial for these graphs. We discuss scenarios with more than 10 layers in Appendix B.

## 6 Conclusion

Our study provides a thorough reevaluation of the efficacy of foundational GNN models in node classification tasks. Through extensive empirical analysis, we demonstrate that these classic GNN models can reach or surpass the performance of GTs on various graph datasets, challenging the perceived superiority of GTs in node classification tasks. Furthermore, our comprehensive ablation studies provide insights into how various GNN configurations impact performance. We hope our findings promote more rigorous empirical evaluations in graph machine learning research.

## Acknowledgments and Disclosure of Funding

We extend our gratitude to Yiwen Sun for her invaluable assistance. We also express our appreciation to all the anonymous reviewers and ACs for their insightful and constructive feedback. This work received support from National Key R&D Program of China (2021YFB3500700), NSFC Grant 62172026, National Social Science Fund of China 22&ZD153, the Fundamental Research Funds for the Central Universities, State Key Laboratory of Complex & Critical Software Environment (CCSE), HK PolyU Grant P0051029, HK PolyU Grant P0038850, and HK ITF Grant ITS/359/21FP. Lei Shi is with Beihang University and State Key Laboratory of Complex & Critical Software Environment.

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

# A   Datasets and Experimental Details

## A.1   Computing Environment

Our implementation is based on PyG [16] and DGL [72]. The experiments are conducted on a single workstation with 8 RTX 3090 GPUs. Notably, only the experiments on the pokec dataset are performed on a separate workstation with 2 A100 GPUs.

## A.2   Hyperparameters and Reproducibility

For the hyperparameter selections of classic GNNs, in addition to what we have covered, we list other settings in Tables 8, 9, 10. Notably, for heterophilous graphs, we expand the search range for the number of layers to include three additional settings: $\{12, 15, 20\}$ (See Section B.2 for further analysis). This adjustment is based on our empirical evidence suggesting that deep networks tend to yield performance improvements on heterophilous graphs. The ReLU function serves as the non-linear activation. Further details regarding hyperparameters can be found in our code https://github.com/LUOyk1999/tunedGNN.

For hyperparameters specific to each GT, which are not present in classic GNNs, we tuned them according to the search space specified in the original GT papers:

- GraphGPS: the number of heads from $\{1, 2, 4\}$, GNNs from $\{GCN, GraphSAGE, GAT\}$, positional encoding schemes from $\{None, LapPE, RWSE\}$.
- NAGphormer: the number of heads from $\{1, 2, 4\}$, number of hops from $\{3, 10\}$.
- Exphormer: the number of heads from $\{1, 2, 4\}$, positional encoding schemes from $\{None, LapPE, RWSE\}$.
- GOAT: the number of heads from $\{1, 2, 4\}$, codebook size from $\{1024, 2048, 4096\}$.
- NodeFormer: the number of heads from $\{1, 2, 4\}$, $M$ from $\{30, 50\}$, $K$ from $\{5, 10\}$, rb from $\{1, 2\}$, temperature from $\{0.25\}$.
- Polynormer: the number of heads from $\{1, 2, 4, 8\}$.
- SGFormer: the number of heads from $\{1, 2, 4\}$, GNN weight from $\{0.5, 0.8\}$.

Due to the large size of the graphs in ogbn-proteins, ogbn-products, and pokec, which prevents full-batch training on GPU memory, we adopt different batch training strategies. For ogbn-proteins, we utilize the optimized neighbor sampling method [20]. For pokec and ogbn-products, we apply the random partitioning method previously used by GTs [12, 76, 75] to enable mini-batch training. For other datasets, we employ full-batch training.

The testing accuracy achieved by the model that reports the highest result on the validation set is used for evaluation. Additionally, we report mean scores and standard deviations after 5 independent runs with different initializations.

Our code is available under the MIT License.

# B   Additional Benchmarking Results

## B.1   GAT* with Edge Features on ogbn-proteins

While DeepGCN [33] introduced training models up to 56 layers deep and DeeperGCN [34] further extended this to 112 layers, our experiments suggest that such depth is not necessary. Specifically, while the DeeperGCN achieved an accuracy of 85.50% on ogbn-proteins, it utilized edge features as input, a configuration not commonly employed in the standard baselines of the OGB dataset [24]. As our experiments do not incorporate edge features on ogbn-proteins, we exclude DeeperGCN from the main text to maintain a fair comparison.

Table 8: Dataset-specific hyperparameter settings of GCN*.

| Dataset | ResNet | Normalization | Dropout rate | GNNs layer $L$ | Hidden dim | LR | epoch |
|---|---|---|---|---|---|---|---|
| Cora | False | False | 0.7 | 3 | 512 | 0.001 | 500 |
| Citeseer | False | False | 0.5 | 2 | 512 | 0.001 | 500 |
| Pubmed | False | False | 0.7 | 2 | 256 | 0.005 | 500 |
| Computer | False | LN | 0.5 | 3 | 512 | 0.001 | 1000 |
| Photo | True | LN | 0.5 | 6 | 256 | 0.001 | 1000 |
| CS | True | LN | 0.3 | 2 | 512 | 0.001 | 1500 |
| Physics | True | LN | 0.3 | 2 | 64 | 0.001 | 1500 |
| WikiCS | False | LN | 0.5 | 3 | 256 | 0.001 | 1000 |
| Squirrel | True | BN | 0.7 | 4 | 256 | 0.01 | 500 |
| Chameleon | False | False | 0.2 | 5 | 512 | 0.005 | 200 |
| Amazon-Ratings | True | BN | 0.5 | 4 | 512 | 0.001 | 2500 |
| Roman-Empire | True | BN | 0.5 | 9 | 512 | 0.001 | 2500 |
| Minesweeper | True | BN | 0.2 | 12 | 64 | 0.01 | 2000 |
| Questions | True | False | 0.3 | 10 | 512 | 0.001 | 1500 |
| ogbn-proteins | True | BN | 0.3 | 3 | 512 | 0.01 | 100 |
| ogbn-arxiv | True | BN | 0.5 | 5 | 512 | 0.0005 | 2000 |
| ogbn-products | False | LN | 0.5 | 5 | 256 | 0.003 | 300 |
| pokec | True | BN | 0.2 | 7 | 256 | 0.0005 | 2000 |

Table 9: Dataset-specific hyperparameter settings of GraphSAGE*.

| Dataset | ResNet | Normalization | Dropout rate | GNNs layer $L$ | Hidden dim | LR | epoch |
|---|---|---|---|---|---|---|---|
| Cora | False | False | 0.7 | 3 | 256 | 0.001 | 500 |
| Citeseer | False | False | 0.2 | 3 | 512 | 0.001 | 500 |
| Pubmed | False | False | 0.7 | 4 | 512 | 0.005 | 500 |
| Computer | False | LN | 0.3 | 4 | 64 | 0.001 | 1000 |
| Photo | True | LN | 0.2 | 6 | 64 | 0.001 | 1000 |
| CS | True | LN | 0.5 | 2 | 512 | 0.001 | 1500 |
| Physics | True | BN | 0.7 | 2 | 64 | 0.001 | 1500 |
| WikiCS | False | LN | 0.7 | 2 | 256 | 0.001 | 1000 |
| Squirrel | True | BN | 0.7 | 3 | 256 | 0.01 | 500 |
| Chameleon | True | BN | 0.7 | 4 | 256 | 0.01 | 200 |
| Amazon-Ratings | True | BN | 0.5 | 9 | 512 | 0.001 | 2500 |
| Roman-Empire | False | BN | 0.3 | 9 | 256 | 0.001 | 2500 |
| Minesweeper | True | BN | 0.2 | 15 | 64 | 0.01 | 2000 |
| Questions | False | LN | 0.2 | 6 | 512 | 0.001 | 1500 |
| ogbn-proteins | True | BN | 0.3 | 6 | 512 | 0.01 | 1000 |
| ogbn-arxiv | True | BN | 0.5 | 4 | 256 | 0.0005 | 2000 |
| ogbn-products | False | LN | 0.5 | 5 | 256 | 0.003 | 1000 |
| pokec | True | BN | 0.2 | 7 | 256 | 0.0005 | 2000 |

Table 10: Dataset-specific hyperparameter settings of GAT*.

| Dataset | ResNet | Normalization | Dropout rate | GNNs layer $L$ | Hidden dim | LR | epoch |
|---|---|---|---|---|---|---|---|
| Cora | True | False | 0.2 | 3 | 512 | 0.001 | 500 |
| Citeseer | True | False | 0.5 | 3 | 256 | 0.001 | 500 |
| Pubmed | False | False | 0.5 | 2 | 512 | 0.01 | 500 |
| Computer | False | LN | 0.5 | 2 | 64 | 0.001 | 1000 |
| Photo | True | LN | 0.5 | 3 | 64 | 0.001 | 1000 |
| CS | True | LN | 0.3 | 1 | 256 | 0.001 | 1500 |
| Physics | True | BN | 0.7 | 2 | 256 | 0.001 | 1500 |
| WikiCS | True | LN | 0.7 | 2 | 512 | 0.001 | 1000 |
| Squirrel | True | BN | 0.5 | 7 | 512 | 0.005 | 500 |
| Chameleon | True | BN | 0.7 | 2 | 256 | 0.01 | 200 |
| Amazon-Ratings | True | BN | 0.5 | 4 | 512 | 0.001 | 2500 |
| Roman-Empire | True | BN | 0.3 | 10 | 512 | 0.001 | 2500 |
| Minesweeper | True | BN | 0.2 | 15 | 64 | 0.01 | 2000 |
| Questions | True | LN | 0.2 | 3 | 512 | 0.001 | 1500 |
| ogbn-proteins | True | BN | 0.3 | 7 | 512 | 0.01 | 1000 |
| ogbn-arxiv | True | BN | 0.5 | 5 | 256 | 0.0005 | 2000 |
| ogbn-products | False | LN | 0.5 | 5 | 256 | 0.003 | 1000 |
| pokec | True | BN | 0.2 | 7 | 256 | 0.0005 | 2000 |

Table 11: Node classification results on ogbn-proteins (%).

| | ogbn-proteins |
|---|---|
| Metric | ROC-AUC↑ |
| DeeperGCN | 85.80 ± 0.17 |
| **GAT**[*] (with edge features) | 87.82 ± 0.16 |

Table 12: Ablation study of the number of layers $L$ on heterophilous graphs (%).

| | Roman-Empire | Minesweeper |
|---|---|---|
| Metric | Accuracy↑ | ROC-AUC↑ |
| **GCN**[*] ($L = 12$) | 90.68 ± 0.44 | 97.76 ± 0.24 |
| **GCN**[*] ($L = 15$) | 90.74 ± 0.38 | 97.65 ± 0.81 |
| **GCN**[*] ($L = 20$) | 90.43 ± 0.52 | 97.52 ± 0.28 |
| **GrapSAGE**[*] ($L = 12$) | 90.96 ± 0.46 | 97.02 ± 0.51 |
| **GrapSAGE**[*] ($L = 15$) | 90.78 ± 0.63 | 97.77 ± 0.62 |
| **GrapSAGE**[*] ($L = 20$) | 90.22 ± 0.69 | 97.73 ± 0.88 |

Table 13: Node classification results over homophilous graphs (%). [+] indicates the implementation of classic GNNs using JK as a hyperparameter configuration in our past experiments.

| | Cora | CiteSeer | PubMed | Computer | Photo | CS | Physics | WikiCS |
|---|---|---|---|---|---|---|---|---|
| Metric | Accuracy↑ | Accuracy↑ | Accuracy↑ | Accuracy↑ | Accuracy↑ | Accuracy↑ | Accuracy↑ | Accuracy↑ |
| **GCN**[+] | 85.08 ± 0.52 | 72.98 ± 0.84 | 81.32 ± 0.72 | 93.80 ± 0.29 | 96.51 ± 0.20 | 95.80 ± 0.28 | 97.43 ± 0.05 | 80.27 ± 0.71 |
| **GraphSAGE**[+] | 84.18 ± 0.81 | 71.93 ± 0.85 | 79.41 ± 0.53 | 93.59 ± 0.22 | 96.41 ± 0.17 | 96.12 ± 0.24 | 97.21 ± 0.05 | 80.51 ± 0.48 |
| **GAT**[+] | 84.64 ± 1.27 | 72.10 ± 1.10 | 79.70 ± 0.70 | 93.93 ± 0.16 | 96.67 ± 0.13 | 96.08 ± 0.10 | 97.30 ± 0.06 | 80.75 ± 0.74 |

Table 14: Node classification results on heterophilous graphs (%).

| | Squirrel | Chameleon | Amazon-Ratings | Roman-Empire | Minesweeper | Questions |
|---|---|---|---|---|---|---|
| Metric | Accuracy↑ | Accuracy↑ | Accuracy↑ | Accuracy↑ | ROC-AUC↑ | ROC-AUC↑ |
| **GCN**[+] | 44.50 ± 1.92 | 46.11 ± 3.16 | 53.57 ± 0.32 | 91.35 ± 0.37 | 97.77 ± 0.38 | 77.40 ± 1.07 |
| **GraphSAGE**[+] | 39.93 ± 1.58 | 43.44 ± 3.19 | 54.72 ± 0.38 | 92.19 ± 0.58 | 96.95 ± 0.41 | 77.96 ± 0.72 |
| **GAT**[+] | 38.72 ± 1.46 | 43.44 ± 3.00 | 54.99 ± 0.71 | 91.60 ± 0.21 | 97.76 ± 0.37 | 79.04 ± 1.27 |

Table 15: Node classification results on large-scale graphs (%).

| | ogbn-proteins | ogbn-arxiv | ogbn-products | pokec |
|---|---|---|---|---|
| Metric | ROC-AUC↑ | Accuracy↑ | Accuracy↑ | Accuracy↑ |
| **GCN**[+] | 77.29 ± 0.46 | 73.60 ± 0.18 | 82.33 ± 0.19 | 86.33 ± 0.17 |
| **GraphSAGE**[+] | 82.21 ± 0.32 | 72.95 ± 0.31 | 83.89 ± 0.36 | 85.97 ± 0.21 |
| **GAT**[+] | 85.01 ± 0.46 | 73.30 ± 0.16 | 80.99 ± 0.16 | 86.19 ± 0.23 |

Now we incorporate edge features into the GAT[*], same as the approach in [73], with the results shown in Table 11. A 6-layer GAT achieve an accuracy of 87.47%, significantly surpassing the 85.50% by DeeperGCN. This demonstrates that GNNs do not need to be as deep as proposed by DeeperGCN; a range of 2 to 10 layers is typically sufficient.

## B.2 Deeper Networks on Heterophilous Graphs

On heterophilous graphs, the performance of classic GNNs improves with an increasing number of layers limited to 10, as evidenced by Figure 1 in the main text. We explore scenarios with more than 10 layers in this subsection. Specifically, we consider GCN[*] and GraphSAGE[*] with layer configurations of 12, 15, and 20 for the Roman-Empire and Minesweeper datasets. The results are shown in Table 12. The variation in the optimal number of layers ($L$) could stem from the distinct structures inherent in different graphs. Heterophilous graphs may have more complex structures, thus necessitating a higher $L$. However, the slight improvements observed with larger $L$ values suggest

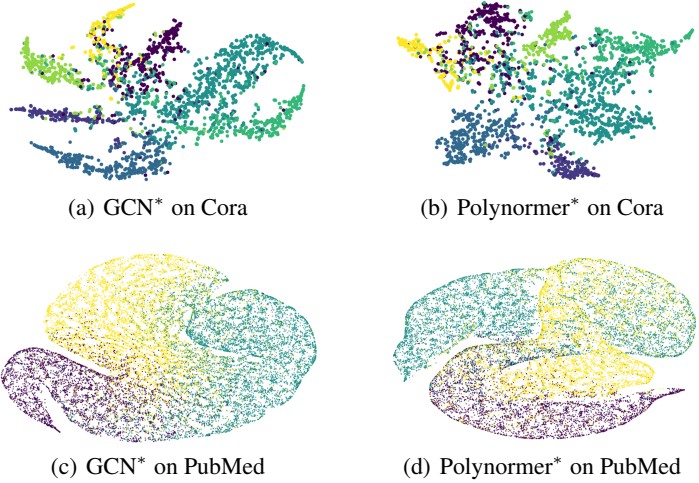

| (a) GCN* on Cora | (b) Polynormer* on Cora |
| --- | --- |
| (c) GCN* on PubMed | (d) Polynormer* on PubMed |

Figure 2: t-SNE visualizations of node embeddings.

that very deep networks may not yield significantly better results. Overall, the best results for classic GNNs are achieved when $L$ is limited to 15.

### B.3 Jumping Knowledge Mode and Early Results

**Jumping Knowledge (JK) Mode** [81] aggregates representations from different GNN layers, effectively capturing information from varying neighborhood ranges within the graph. For any node $v$, the summation version of JK mode produces the representation of $v$ by:

$$\text{GNN}_{\text{JK}}(v, \boldsymbol{A}, \boldsymbol{X}) = \boldsymbol{h}_v^1 + \boldsymbol{h}_v^2 + \ldots + \boldsymbol{h}_v^L, \tag{8}$$

where $L$ is the number of GNN layers. In our previous experimental setups, we treated JK as a hyperparameter configuration for GNNs. Based on the hyperparameter configurations outlined in Section 3, we expanded the tuning space to include the decision of whether to use JK. In past experiments, we did not perform an exhaustive search; instead, we selected subsets based on experience within this search space, and our early results are reported in Table 13, 14, and 15 (For additional information, please refer to https://arxiv.org/abs/2406.08993v1). However, after a more detailed hyperparameter tuning, we found that JK may not be necessary. In most datasets, the results without using JK are comparable to, and sometimes even better than, those with JK. Consequently, we removed JK from the hyperparameter tuning search space in our paper.

## C  Visualization

Here, we present t-SNE visualizations of classification results. As shown in Figure 2, the node embeddings generated by GCN* (our implementation) display greater inter-class distances than those produced by Polynormer*.

## D  Limitations & Broader Impacts

**Broader Impacts.** This paper presents work whose goal is to advance the field of Machine Learning. There are many potential societal consequences of our work, none which we feel must be specifically highlighted here.

**Limitations.** In this study, we focus solely on the node classification task, without delving into graph classification [14, 44, 43] and link prediction [39, 87] tasks. It would be beneficial to extend our benchmarking efforts to include classic GNNs in graph-level and edge-level tasks.

