# OpenReview forum: "Classic GNNs are Strong Baselines: Reassessing GNNs for Node Classification"
_NeurIPS.cc/2024/Datasets_and_Benchmarks_Track — NeurIPS 2024 Track Datasets and Benchmarks Poster_

### Official Review · Reviewer_madU · 2024-07-19
**This work is emperically convincing since the experiments are sufficient, but whether this is theoretically true requires further research.**

**Rating:** 8
**Confidence:** 3
**Correctness:** Yes, the claims made in the submissio…
**Clarity:** Yes, the paper is written well.

**Review:**

### Pros
1. This paper uses extensive experimental results to show empirically that GTs are overstated in node classification.
2. Sufficient ablation studies give valuable insights into GNN hyperparameters.
3. Many empirical conclusions in hyperparameter tuning are of reference significance.
### Cons
1. All the results are numerical. It would be better for reading if there were some visualizations.

**Strengths:**

This work uses extensive experimental results to show that classic GNNs can achieve highly competitive performance in node classification with proper hyperparameter tuning. Moreover, the authors demonstrate some important insights into normalization, dropout, residual connection, and layer depth.

**Additional Feedback:**

### Questions
1. For large-scale graphs, if we use the mini-batch strategy, will GNNs outperform GTs?
2. is the hyperparameter tuning for GTs proper like it is in classic GNNs?

**Documentation:**

Yes, the documentation is sufficient.

**Ethics:**

There are no ethical concerns.

**Limitations:**

The authors should discuss limitations in the paper, e.g., some potential unexpected circumstances.

**Opportunities For Improvement:**

Given some visualizations of classification results for GNNs and GTs.

**Relation To Prior Work:**

No, this paper does not discuss how this work differs from previous contributions.

**Summary And Contributions:**

This work reassesses three classic GNNs in the node classification task empirically and finds that these GNNs can reach SOTA after hyperparameters tuning. Extensive experiments yield valuable insights into GNN hyperparameters.

---

> ### Author Rebuttal · Authors · 2024-08-17
>
> **Thank you very much for fully recognizing our contributions!**
>
> **Comment 1: Visualization Results**
>
> > All the results are numerical. It would be better for reading if there were some visualizations.
> >
> > Given some visualizations of classification results for GNNs and GTs.
>
> Thank you for the suggestion. Following the advice, we add some t-SNE visualizations of classification results in Figure 1 of the attached PDF. We observe that the node embeddings generated by GCN* (our implementation) exhibit greater inter-class distances compared to those generated by Polynormer. We will incorporate the results in the revised version.
>
> **Question 1: Mini-Batch Training Strategy for Large-scale Graphs**
>
> > For large-scale graphs, if we use the mini-batch strategy, will GNNs outperform GTs?
>
> Yes, as stated in lines 13-16 of Appendix A.2 (in Supplementary Material), we employed a mini-batch training strategy for large-scale graphs, consistent with approaches used in GTs like SGformer and Polynormer. The results demonstrate that the classic GNNs outperform these GTs under the mini-batch training strategy.
>
> **Question 2: Hyperparameter Tuning for GTs**
>
> > Is the hyperparameter tuning for GTs proper like it is in classic GNNs?
>
> Thank you for the insightful question! Please refer to our golabl rebuttal.
>
> Thank you once again for helping us improve our work!

---

> > ### Comment · Reviewer_madU · 2024-08-18
> >
> > Thanks for your response, my concerns are addressed. I think this paper is good work and I will raise the score.

---

> > > ### Author Response · Authors · 2024-08-18
> > >
> > > Dear Reviewer madU,
> > >
> > > We sincerely appreciate the time and effort you have dedicated to reviewing our response and for raising your score. Your recognition of our work is highly valued and means a great deal to us. Thank you very much.
> > >
> > > Best regards,
> > >
> > > Authors

---

### Official Review · Reviewer_PM1d · 2024-07-23
**Problematic experimental setup invalidates claims. Good ablation study.**

**Rating:** 5
**Confidence:** 5
**Correctness:** Problematic experimental design
**Clarity:** Yes

**Review:**

**List of pros**
* Ablation study
* Presentation
* Dataset variety

**List of cons**
* Problematic experimental setup

**Strengths:**

1. The ablation study highlighting where individual GNN configurations shine is very insightful for GNN training. In addition to highlighting these points, the authors provide reasoning behind them as well.
2. The paper is well-written and well-presented. It is easy to follow.
3. The authors have curated a good mix of datasets covering heterophilous and large-scale datasets.

**Additional Feedback:**

* In table 1, the number of features for Cora is incorrect. It should be 1433.
* In table 1, the number of edges has been halved for homophilous datasets but not for heterophilous datasets. I believe both datasets are undirected.
* In equation 7, $Norm(h_u)$ should be replaced with $Norm(h_v)$.

**Documentation:**

The authors have provided a codebase.

**Limitations:**

Yes

**Opportunities For Improvement:**

The authors have fallen into the second pitfall highlighted by “Shchur, Oleksandr, et al. "Pitfalls of graph neural network evaluation." arXiv preprint arXiv:1811.05868 (2018).” which states: “Second, when evaluating performance of a new model, people often use a training procedure that is rather different from the one used for the baselines. This makes it difficult to identify whether the improved performance comes from (a) a superior architecture of the new model, or (b) a better-tuned training procedure and / or hyperparameter configuration that unfairly benefits the new model.”

The authors report metrics from other papers and train their models separately. This leads to different training environments and an unfair comparison with the baselines rendering any conclusions drawn henceforth against transformers and classical GNNs moot. The authors report their results from these three papers:

[1] Chenhui Deng, Zichao Yue, and Zhiru Zhang. Polynormer: Polynomial-expressive graph transformer in linear time. arXiv preprint arXiv:2403.01232, 2024.

[2] Oleg Platonov, Denis Kuznedelev, Michael Diskin, Artem Babenko, and Liudmila Prokhorenkova. A critical look at the evaluation of gnns under heterophily: Are we really making progress? arXiv preprint arXiv:2302.11640, 2023.

[3] Qitian Wu, Wentao Zhao, Chenxiao Yang, Hengrui Zhang, Fan Nie, Haitian Jiang, Yatao Bian, and Junchi Yan. Simplifying and empowering transformers for large-graph representations. In Thirty-seventh Conference on Neural Information Processing Systems, 2023.

[1] commits the same mistake as the authors of reporting baseline metrics from prior work while training their own models separately. However, the authors should take note that both [2, 3] retrain their baseline classical GNNs to ensure an identical training environment resulting in a fair comparison. The authors are expected to emulate the same.


With regards to the ablation, the results are averaged over 5 runs. It is not clear to me what a “run” means here. Do two runs differ in the train/val/test split; the initialization seeds of the models; or both?

**Relation To Prior Work:**

NA

**Summary And Contributions:**

This study claims that Graph Transformers under-report their classical GNN baselines due to poor hyperparameter tuning. The authors begin by reporting metrics of Transformers and classical GNNs from prior studies. They then proceed to tune the hyper-parameters of the classical GNNs. The results show that the tuned classical GNNs outperform Transformers on 17 out of 18 datasets, including homophilous, heterophilous, and large-scale datasets. Additionally, the study contributes an ablation analysis of various GNN configurations, such as normalization and residual connections. This analysis provides fruitful insights and reasoning, indicating that certain configurations are more critical to specific subdomains. For instance, residual connections prove to be more advantageous for heterophilous graphs compared to others.

---

> ### Author Rebuttal · Authors · 2024-08-17
>
> **We greatly appreciate your comprehensive understanding of our work and your recognition of our contributions. We hope our response below effectively addresses your concerns.**
>
> **Comment 1: Concerns on Different Training Environments**
>
> > The authors report metrics from other papers and train their models separately. This leads to different training environments and an unfair comparison with the baselines rendering any conclusions drawn henceforth against transformers and classical GNNs moot.
> >
> > The authors should retrain their baseline to ensure an identical training environment resulting in a fair comparison. The authors are expected to emulate the same.
>
> We appreciate your insightful comment! Please refer to our golabl rebuttal.
>
> **Comment 2: Clarification on the Meaning of "Run"**
>
> > With regards to the ablation, the results are averaged over 5 runs. It is not clear to me what a “run” means here. Do two runs differ in the train/val/test split; the initialization seeds of the models; or both?
>
> We apologize for the lack of clarity.
>
> In this context, each run refers to an independent trial with a **different initialization seed**, while using the **same standard train/val/test split** as detailed in lines 129-154 of our manuscript. For each experiment, we conducted 5 such independent trials and reported the mean and variance of the performance metrics. This approach aligns with standard practice and is consistent with the evaluation method used in studies such as SGformer and Polynormer.
>
> **Regarding "Additional feedback"**
>
> > In table 1, the number of features for Cora is incorrect. It should be 1433.
> >
> > In equation 7, $Norm(h_u)$ should be replaced with $Norm(h_v)$.
>
> We sincerely appreciate your careful reading! We will update the information and correct the typographical errors in the revised version.
>
> > In table 1, the number of edges has been halved for homophilous datasets but not for heterophilous datasets. I believe both datasets are undirected.
>
> Regarding the number of edges reported in Table 1, we have checked that **the statistics in our manuscript are accurate**; the number of edges reflects the original edge sizes without doubling for conversion to undirected edges. This consistency applies to both homophilous and heterophilous datasets. A review of our code revealed that while printing statistics for homophilous datasets, the number of edges was inadvertently doubled, which did not occur for heterophilous datasets. This likely caused the misunderstanding. We will update our code to correct this issue.
>
> Thank you once again for helping us improve our work!

---

> > ### Comment · Reviewer_PM1d · 2024-08-20
> >
> > I thank the authors for the clarifications. I would like to stick to a rating of 5, since I believe it is unacceptable to directly pick numbers from a different paper. The initialization seeds, train-test splits, and many other factors would vary. Even the GPU and CUDA environments are known to alter results. In a benchmarking paper, these variables need to be eliminated. I would expect the authors to (i) conduct fresh experiments with identical seeds, splits, environments, etc. Furthermore, all trained models should be released for full reproducibility (and not just the code).

---

> > ### Author Response · Authors · 2024-08-20
> >
> > Dear Reviewer PM1d,
> >
> > Thank you for your further feedback!
> >
> > We are unsure if you have had the opportunity to read our global rebuttal, where **we have specifically addressed the questions you are most concerned about**. If you have not yet reviewed it, please note that we have provided **a new global rebuttal with fully updated results of the GTs, in accordance with your suggestions**.
> >
> > Specifically, we retrained all GTs (GraphGPS, NAGphormer, Exphormer, GOAT, NodeFormer, SGFormer, and Polynormer) using the same **hyperparameter tuning search space** and **training environments** as the classic GNNs. The "same training environments" include using **identical initialization seeds** (0, 1, 2, 42, 123), **train/val/test splits** (lines 129-154), and **GPU environments** (GPU: RTX 3090 GPUs, CUDA Version: 11.3, Python 3.7, PyTorch 1.12.1), with the only difference being the model architectures themselves.
> >
> >
> > For your convenience, we also provide the updated results in the table below, with the **complete results available in Tables 1-3 in the PDF attached to the new global rebuttal**.
> >
> > |                 | Cora↑            | CiteSeer↑        | PubMed↑          | Computer↑        | Photo↑           | CS↑              | Physics↑         | WikiCS↑          | ogbn-proteins↑   | ogbn-arxiv↑      | ogbn-products↑   | pokec↑           |
> > | --------------- | ---------------- | ---------------- | ---------------- | ---------------- | ---------------- | ---------------- | ---------------- | ---------------- | ---------------- | ---------------- | ---------------- | ---------------- |
> > | **GraphGPS***   | 83.87 ± 0.96     | 72.73 ± 1.23     | 79.94 ± 0.26     | 91.79 ± 0.63     | 94.89 ± 0.14     | 94.04 ± 0.21     | 96.71 ± 0.15     | 78.66 ± 0.49     | 77.15 ± 0.64     | 71.23 ± 0.59     | OOM              | OOM              |
> > | **NodeFormer*** | 82.73 ± 0.75     | 72.37 ± 1.20     | 79.59 ± 0.92     | 87.29 ± 0.58     | 93.43 ± 0.56     | 95.69 ± 0.27     | 96.48 ± 0.34     | 75.13 ± 0.93     | 77.86 ± 0.84     | 67.78 ± 0.28     | 73.96 ± 0.30     | 71.00 ± 1.30     |
> > | **SGFormer***   | 84.82 ± 0.85     | 72.72 ± 1.15     | 80.60 ± 0.49     | 92.42 ± 0.66     | 95.58 ± 0.36     | 95.71 ± 0.24     | 96.75 ± 0.26     | 80.05 ± 0.46     | 79.92 ± 0.48     | 72.76 ± 0.33     | 81.54 ± 0.43     | 82.44 ± 0.76     |
> > | **Polynormer*** | 83.43 ± 0.89     | 72.19 ± 0.83     | 79.35 ± 0.73     | 93.78 ± 0.10     | 96.57 ± 0.23     | 95.42 ± 0.19     | 97.18 ± 0.11     | 80.26 ± 0.92     | 79.53 ± 0.67     | 73.40 ± 0.22     | 83.82 ± 0.11     | 86.06 ± 0.25     |
> > | **NAGphormer*** | 80.92 ± 1.17     | 70.59 ± 0.89     | 80.14 ± 1.06     | 91.69 ± 0.30     | 96.14 ± 0.16     | 95.85 ± 0.16     | 97.35 ± 0.12     | 77.92 ± 0.93     | 72.17 ± 0.45     | 70.88 ± 0.24     | 74.63 ± 0.29     | 75.92 ± 0.68     |
> > | **Exphormer***  | 83.29 ± 1.36     | 71.85 ± 1.11     | 79.67 ± 0.73     | 91.80 ± 0.35     | 95.69 ± 0.39     | 95.92 ± 0.25     | 97.06 ± 0.13     | 79.38 ± 0.62     | 77.62 ± 0.33     | 72.32 ± 0.36     | OOM              | OOM              |
> > | **GOAT***       | 83.26 ± 1.24     | 72.21 ± 1.29     | 80.06 ± 0.67     | 92.29 ± 0.37     | 94.33 ± 0.21     | 93.81 ± 0.19     | 96.47 ± 0.16     | 77.96 ± 0.63     | 79.31 ± 0.42     | 72.76 ± 0.29     | 82.27 ± 0.56     | 72.64 ± 0.67     |
> > | **GCN***        | **85.08 ± 0.52** | **72.98 ± 0.84** | **81.32 ± 0.72** | 93.80 ± 0.29     | 96.51 ± 0.20     | 95.80 ± 0.28     | **97.43 ± 0.05** | 80.27 ± 0.71     | 77.29 ± 0.46     | **73.60 ± 0.18** | 82.33 ± 0.19     | **86.33 ± 0.17** |
> > | **GraphSAGE***  | 84.18 ± 0.81     | 71.93 ± 0.85     | 79.41 ± 0.53     | 93.59 ± 0.22     | 96.41 ± 0.17     | **96.12 ± 0.24** | 97.21 ± 0.05     | 80.51 ± 0.48     | 82.21 ± 0.32     | 72.95 ± 0.31     | **83.89 ± 0.36** | 85.97 ± 0.21     |
> > | **GAT***        | 84.64 ± 1.27     | 72.10 ± 1.10     | 79.70 ± 0.70     | **93.93 ± 0.16** | **96.67 ± 0.13** | 96.08 ± 0.10     | 97.30 ± 0.06     | **80.75 ± 0.74** | **85.01 ± 0.46** | 73.30 ± 0.16     | 80.99 ± 0.16     | 86.19 ± 0.23     |
> >
> >
> > The **updated results do not alter our conclusion** that classic GNNs can achieve performance that is competitive with, or even superior to, that of GTs. To **ensure the reproducibility of our findings on the classic GNNs**, we have included the **source code in the supplementary material** submitted with the manuscript. We are committed to releasing all the code and trained models on a public GitHub repository upon the acceptance of our paper.
> >
> > We believe we have thoroughly addressed your concerns and kindly hope you will reassess our work in light of the updated results.
> >
> > Best Regards,
> >
> > Authors

---

> > > ### Comment · Reviewer_PM1d · 2024-08-21
> > >
> > > I did read the global rebuttal. However, it did not cover all of the GTs. Hence, I am not confident if the claim made by the authors is sound.
> > >
> > > Looking at the new results posted on 20th August, I still have several questions.
> > > 1. Can you present the train and validation losses on the GTs to ensure they are well trained?
> > > 2. How did you select the number of layers in each GT? GTs do not show the effects of over-smoothing and over-squashing [1]. Can you produce plots showing the effect of number of GT layers with accuracy?
> > > 3. You have nice tables in the appendix where you show the optimal parameters for GCN, GAT and GraphSage. The same tables with the same level of detail should be produced for each of the benchmarked GTs. In GPS for example, various MPNNs and positional encoding schemes can be used. Did you try to optimize for all?
> > > 4. GPS and Exphormer use MPNNs as well as positional encodings. If you replace this MPNN within GPS or Exphormer with the best performing MPNN, do they still continue to outperform? If yes, can you offer insights into such trends?
> > > 5. Are the results statistically significant? For instance, in CORA, GCN achieves 72.98 ± 0.84 compared to 72.72 ± 1.15 by SGFormer and 72.73 ± 1.23 by GPS.
> > >
> > > Finally, given the empirical nature of this paper, I would request that you release the model weights, codes, and config files through an anonymous repo. It cannot be conditional on acceptance. I would like to reproduce at least some of the experiments on my end to ensure that the claims are sound.
> > >
> > > [1] Do Transformer Really Perform Badly for Graph Representation Learning? NeurIPS 2021.

---

> > > > ### Author Response · Authors · 2024-08-23
> > > >
> > > > We appreciate your further feedback.
> > > >
> > > > It is important to note that
> > > >
> > > > **(1)** The results of GTs reported in our manuscript and the PDF in our rebuttal (whichever is better per case) represent **the best results obtained on these datasets with the standard splits**, to the best of our knowledge. We have not encountered better results reported in existing papers or leaderboards.
> > > >
> > > > **(2)** ***Our study is NOT about benchmarking GTs.*** Instead, we **focus on assessing classic GNNs on node classification** tasks. **We have ensured a fair comparision between GNNs and GTs** by using the same hyperparameter search space (for their shared hyperparameters, see details in our previous global responses) and training environments. Our findings show that the **performance of GNNs was significantly underestimated** in previous GT papers, and GNNs are capable of achieving comparable or even better performance than state-of-the-art GTs.
> > > >
> > > > Therefore, we believe **we have performed our due diligence**, and **exhaustively tuning various GTs with their respective hyperparameters is beyond the scope of this paper**. In fact, given the large and diverse set of hyperparameters in some GTs, such as GraphGPS mentioned by you, it is very challenging, if not impossible, to achieve this.
> > > >
> > > > Questions:
> > > >
> > > > > I would request that you release the model weights, codes, and config files through an anonymous repo.
> > > >
> > > > We have included the source code in the supplementary material upon submission of the manuscript to ensure the reproducibility of our results on the classic GNNs. We are currently checking with the ACs/PCs if providing an external link is permitted. **If allowed, we will also provide the code and model necessary to reproduce the results on the baseline GTs** through an anonymous repo.
> > > >
> > > > > **Q1:** Can you present the train and validation losses on the GTs to ensure they are well trained?
> > > >
> > > > The results are provided in Figure 1 of the new PDF attached to the new rebuttal in this thread. We present the training performance of all GTs on the Photo dataset, demonstrating that they were well trained. Similar patterns were observed in other datasets, where the losses also converged.
> > > >
> > > > > **Q2(1):** How did you select the number of layers in each GT?
> > > >
> > > > We tune the number of layers in each GT to {1, 2, 3, 4, 5, 6, 7, 8, 9, 10}, same as we did for the classic GNNs.
> > > >
> > > > > **Q2(2):** GTs do not show the effects of over-smoothing and over-squashing [1]. Can you produce plots showing the effect of number of GT layers with accuracy?
> > > >
> > > > The results are provided in Figure 2 of the new PDF attached to the new rebuttal in this thread. It displays a trend that as the number of layers increases, the performance of GTs tends to worsen. This observation is consistent with findings in papers such as GOAT and SGFormer, where the number of GT layers is consistently set to 1 across all datasets. NodeFormer sets the maximum number of layers to 3, while Exphormer sets it to 4. Additionally, the NAGphormer paper mentions in Section 4.5 that a smaller L (number of layers) can achieve high accuracy, while a larger L degrades performance. This requires more theoretical explanation beyond the statement that "GTs do not show the effects of over-smoothing and over-squashing". It is likely that they face other issues, such as difficulty in optimization.
> > > >
> > > > > **Q3:** You have nice tables in the appendix where you show the optimal parameters for GCN, GAT and GraphSage. The same tables with the same level of detail should be produced for each of the benchmarked GTs. In GPS for example, various MPNNs and positional encoding schemes can be used. Did you try to optimize for all?
> > > >
> > > > As aforementioned, exhaustively tuning various GTs with their respective hyperparameters is beyond the scope of this paper.
> > > >
> > > > Additionally, given that each GT has its own set of hyperparameters, it is impractical to create tables like those in the appendix due to the sheer number of hyperparameters. But we can provide the search space for these hyperparameters, which aligns with the configurations presented in their respective papers.
> > > >
> > > > - **GraphGPS:** the number of heads from {1, 2, 4}, MPNNs from {GCN, GraphSAGE, GAT}, positional encoding schemes from {None, LapPE, RWSE}.
> > > > - **NAGphormer:** the number of heads from {1, 2, 4}, the number of hops from {3, 10}.
> > > > - **Exphormer:** the number of heads from {1, 2, 4}, positional encoding schemes from {None, LapPE, RWSE}.
> > > > - **GOAT:** the number of heads from {1, 2, 4}, codebook size {1024, 2048, 4096}.
> > > > - **NodeFormer:** the number of heads from {1, 2, 4}, M from {30, 50}, K from {5, 10}, rb order from {1, 2}, and the temperature from {0.25}.
> > > > - **Polynormer:** the number of heads from {1, 2, 4, 8}.
> > > > - **SGFormer:** the number of heads from {1, 2, 4}, graph weight within {0.5, 0.8}.

---

> > ### Author Response · Authors · 2024-08-23
> >
> > Dear ACs,
> >
> > We are currently unsure if including an external anonymous link is permitted in our rebuttal. Could you please confirm?
> >
> > Thank you so much in advance!
> >
> > Best Regards,
> >
> > Authors

---

> > > ### Author Response · Authors · 2024-08-25
> > >
> > > Dear ACs,
> > >
> > > We previously inquired whether including an external anonymous link, as requested by Reviewer PM1d, is permitted in our rebuttal, but we have not yet received a response. As we are nearing the end of the discussion period, could you please confirm this matter at your earliest convenience? Thank you for your attention.
> > >
> > > Best Regards,
> > >
> > > Authors

---

> > ### Author Response · Authors · 2024-08-23
> >
> > > **Q4:** If you replace this MPNN within GPS or Exphormer with the best performing MPNN, do they still continue to outperform?
> >
> >
> > We replaced the MPNN in GraphGPS with our best performing GCN (GCN with fixed optimized hyperparameters) and then tuned the number of heads and the number of layers for the global attention within the same range as mentioned before to achieve the best results. The results below indicate that GCN still outperforms GraphGPS, suggesting that the optimal GCN configuration, when combined with the global attention mechanism in GraphGPS, may not yield the best performance.
> >
> > |                                           | Computer↑    | Photo↑       | CS↑          | Physics↑     | WikiCS↑      |
> > | ----------------------------------------- | ------------ | ------------ | ------------ | ------------ | ------------ |
> > | **GraphGPS**                              | 91.79 ± 0.63 | 94.89 ± 0.14 | 94.04 ± 0.21 | 96.71 ± 0.15 | 78.66 ± 0.49 |
> > | **GraphGPS with the best-performing GCN** | 90.20 ± 0.88 | 94.35 ± 0.21 | 95.01 ± 0.24 | 97.13 ± 0.10 | 77.30 ± 0.83 |
> > | **GCN**                                   | 93.80 ± 0.29 | 96.51 ± 0.20 | 95.80 ± 0.28 | 97.43 ± 0.05 | 80.27 ± 0.71 |
> >
> >
> > > **Q5:** Are the results statistically significant? For instance, in CORA, GCN achieves 72.98 ± 0.84 compared to 72.72 ± 1.15 by SGFormer and 72.73 ± 1.23 by GPS.
> >
> > The numbers you mentioned are the results for CiteSeer, not CORA.
> >
> > Please note that **we are not claiming that classic GNNs are statistically significantly better than GTs in every dataset**. On the contrary, our results show that **GNNs achieve comparable, and sometimes even better, performance to state-of-the-art GTs** in all 18 datasets tested. This highlights the **substantial gaps** with the results reported in previous GT papers and the **potential of GNNs**.
> >
> > ---------------------------------------------
> >
> > To further address the reviewer’s concerns regarding the training environments, we have designed a new set of experiments using the latest two GTs: SGFormer (NeurIPS 2023), and Polynormer (ICLR 2024). We conducted the experiments following all the settings in their respective papers, including the same model hyperparameter search space, dataset splits, optimizer, initialization, learning rate decay, batch training, maximum number of training epochs, early stopping criterion, patience, and validation frequency.
> >
> > We successfully reproduced their reported results (displayed in the table below) in their training environments. Then, we trained a GCN* in the same environments and report the results below. The performance of the GCN* trained under their conditions shows very small differences compared to our previously reported results (in our paper and PDF rebuttal). This experiment demonstrates that our findings are robust and that the training environments have minimal impact on our experimental results.
> >
> >
> > 1. SGFormer (following all the settings in SGFormer paper)
> >
> >    Hyperparameters:
> >
> >     - Learning Rate: {0.001, 0.005, 0.01, 0.05, 0.1};
> >     - Hidden Dimension: {32, 64, 128, 256};
> >     - Dropout Rates: {0, 0.2, 0.3, 0.5};
> >     - Number of Layers: {1, 2, 3};
> >     - Normalization: {None, BN};
> >     - Residual Connections: Enabled or Disabled.
> >
> >     Training environments:
> >
> >     We adhered strictly to the training environments provided in SGFormer’s code.
> >
> >    |          | Cora↑                | CiteSeer↑            | PubMed↑              | Squirrel↑            | Chameleon↑           |
> >    | :------- | :------------------- | :------------------- | :------------------- | :------------------- | :------------------- |
> >    | SGFormer | 84.50 ± 0.80         | 72.60 ± 0.20         | 80.30 ± 0.60         | 41.80 ± 2.27         | 44.93 ± 3.91         |
> >    | GCN (reported in the SGFormer paper)     | 81.60 ± 0.40         | 71.60 ± 0.40         | 78.80 ± 0.60         | 38.60 ± 1.80         | 41.30 ± 3.00         |
> >    | **GCN*** | 85.08 ± 0.52 (3.48↑) | 72.94 ± 0.40 (1.34↑) | 80.86 ± 0.58 (2.06↑) | 43.08 ± 1.38 (4.48↑) | 45.34 ± 3.19 (4.04↑) |
> >
> > 2. Polynormer (following all the settings in Polynormer paper)
> >
> >    |            | Computer↑            | Photo↑               | CS↑                  | Physics↑             | WikiCS↑              |
> >    | :--------- | :------------------- | :------------------- | :------------------- | :------------------- | :------------------- |
> >    | Polynormer | 93.68 ± 0.21         | 96.46 ± 0.26         | 95.53 ± 0.16         | 97.27 ± 0.08         | 80.10 ± 0.67         |
> >    | GCN (reported in the Polynormer paper)        | 89.65 ± 0.52         | 92.70 ± 0.20         | 92.92 ± 0.12         | 96.18 ± 0.07         | 77.47 ± 0.85         |
> >    | **GCN***   | 93.76 ± 0.08 (4.11↑) | 96.53 ± 0.36 (3.83↑) | 95.91 ± 0.09 (2.99↑) | 97.36 ± 0.03 (1.18↑) | 80.27 ± 0.51 (2.80↑) |

---

> > > ### Author Rebuttal · Authors · 2024-08-23
> > >
> > > We attach a one-page PDF that contains additional results.
> > >
> > > > **Q1:** Can you present the train and validation losses on the GTs to ensure they are well trained?
> > >
> > > The results are provided in Figure 1 of the PDF attached to this new rebuttal.
> > >
> > > > **Q2(2):** GTs do not show the effects of over-smoothing and over-squashing [1]. Can you produce plots showing the effect of number of GT layers with accuracy?
> > >
> > > The results are provided in Figure 2 of the PDF attached to this new rebuttal.

---

> > > > ### Comment · Reviewer_PM1d · 2024-08-25
> > > >
> > > > By your own admission, while you have extensively tuned GNNs, the same has not been done with due diligence on GTs.
> > > >
> > > > > ...exhaustively tuning various GTs with their respective hyperparameters is beyond the scope of this paper.
> > > >
> > > > Hence, the conclusion that can be fairly drawn from this study is that a well-tuned MPNN may outperform sub optimally-tuned GTs. Whether this performance gap is statistically significant, or just a manifestation of randomization is currently missing. This, in my opining, is a significant omission.
> > > >
> > > > This empirical setup makes the claims made in the paper stand on weak/incorrect grounds. For instance:
> > > >
> > > > Line 49:
> > > > > Notably, classic GNNs outperform state-of-the-art GTs, achieving the top rank on 17 out of 18 datasets.
> > > >
> > > > Line 261:
> > > > > we demonstrate that these classic GNN models can reach or surpass the performance of GTs on various graph datasets.
> > > >
> > > > In a benchmarking paper, my expectation would be that when two architectures are being compared, they both should be tuned reasonably well, which is not the case here.
> > > >
> > > > I would also expect insights. Why would GPS not do as well as GCN, when the same GCN is being used an the MPNN component in GPS? I am not saying your results are incorrect, but what insights do I draw so that I (or the community at large) can use while choosing their architecture of choice.
> > > >
> > > > Hence, I feel the conclusions drawn from the numbers presented are not rigorous enough.

---

> > ### Author Response · Authors · 2024-08-27
> >
> > Dear Reviewer PM1d,
> >
> > > While you have extensively tuned GNNs, the same has not been done with due diligence on GTs.
> > > In a benchmarking paper, my expectation would be that when two architectures are being compared, they both should be tuned reasonably well, which is not the case here.
> >
> > About two days ago, we clarified in our response that we **tuned GTs more thoroughly than GNNs**. We are unsure if you had the opportunity to review that clarification.
> >
> > We would like to reiterate that the results we reported for GTs are the **best ever documented** with the standard splits. We believe **this sufficiently demonstrates that the GTs were reasonably tuned**.
> >
> >
> > After reviewing our new experimental results and rebuttal, both Reviewer Ejkm and Reviewer madU have confirmed the fairness and solidity of our resutls.
> >
> > Note that our experiments also include tests on 3 OGB datasets: ogbn-proteins, ogbn-arxiv, and ogbn-products. The OGB datasets have a dedicated website showcasing the OGB leaderboard. The OGB team independently reproduces experiments from papers to ensure their validity. **Our experimental results have been verified by the OGB team that they indeed surpass the results of all GTs on these datasets.**
> >
> > During the rebuttal period, we **made every effort to follow your advice and conducted numerous experiments**. So far, every additional experiment we have conducted has **further confirmed or strengthened our conclusion**. *We find it difficult to understand why you lowered the rating of our work (from 5 to 4), given the substantial new evidence beyond the manuscript that supports our conclusion.*
> >
> > We kindly request that you assess our work based on the facts and evidence we have provided.
> >
> > Best Regards,
> >
> > Authors

---

> > > ### Author Response · Authors · 2024-08-27
> > >
> > > > I would also expect insights. Why would GPS not do as well as GCN, when the same GCN is being used an the MPNN component in GPS? I am not saying your results are incorrect, but what insights do I draw so that I (or the community at large) can use while choosing their architecture of choice.
> > >
> > >
> > > It is **beyond the scope of our paper** to analyze why the best-performing GCN, when used independently, does not perform best when integrated into GraphGPS. The GraphGPS method was originally designed to address graph-level tasks. In fact, the GraphGPS paper does not include node classification tasks on large-scale graphs, nor even on relatively smaller graphs such as Cora.
> > >
> > > Our hypothesis is that the global attention mechanism used in GraphGPS may conflict with the local mechanism of GNNs on large-scale graphs, because it overly focuses on those distant nodes. The weakness of the global attention mechanism in GTs is also identified and experimentally validated by [1], referring to as the over-globalizing problem.
> > >
> > > [1] Less is More: On the Over-Globalizing Problem in Graph Transformer, ICML 2024.

---

> > > > ### Author Response · Authors · 2024-08-30
> > > >
> > > > Dear Reviewer PM1d,
> > > >
> > > > We noticed that you updated your rating of our work back to 5 after reading our response sent on August 27. We are glad that our voice was heard.
> > > >
> > > > We believe we have clearly explained that our experimental comparisons are fair and substantiate our claim, as confirmed by other reviewers. However, if there are any aspects you still have doubts about, please let us know so we can address them.
> > > >
> > > > Since we haven’t received a response from the ACs, we are still unsure if including an external anonymous link is permitted in our rebuttal. We reviewed the NeurIPS 2024 FAQ for Authors, which states that we can send an anonymized link to the AC, as quoted below:
> > > >
> > > > >- **Can we include an anonymous link in the author rebuttal?** No. Do not use links in any part of the response. The only exception is if the reviewers asked for code, in which case you can send an anonymized link to the AC in an Official Comment (make sure all linked files are anonymized).
> > > >
> > > >
> > > > Therefore, we have sent an anonymized link to the ACs via an Official Comment and have requested that they forward it to you. You may also contact the ACs directly to obtain it.
> > > >
> > > > In this link, we have provided source codes that can reproduce the results of the GNNs and GTs reported in our rebuttal. We also prepared all the model files, but we discovered that the anonymous GitHub repository has a default limit of 100MB for individual file uploads. Notably, the models for a single dataset, such as Computer, exceed 2GB in size. Therefore, we have only provided the GCN model trained for the Computer dataset as an example. Our provided codes ensure both the models and results can be reproduced.
> > > >
> > > > Thank you for your time and consideration.
> > > >
> > > > Best Regards,
> > > >
> > > > Authors

---

> ### Author Response · Authors · 2024-08-25
>
> We respectfully disagree with Reviewer PM1d's assertion that "while you have extensively tuned GNNs, the same has not been done with due diligence on GTs." **The hyperparameter tuning range for GTs was significantly larger than that for GNNs.**
>
> Please note that:
>
> 1. We retrained all GTs including GraphGPS, NAGphormer, Exphormer, GOAT, NodeFormer, SGFormer, and Polynormer using the same **hyperparameter tuning search space** and training environments as the classic GNNs.
>
>    The **same hyperparameter tuning search space** including:
>
>    - **Learning Rate:** {0.001, 0.005, 0.01}
>    - **Hidden Dimension:** {64, 256, 512}
>    - **Dropout Rates:** {0.0, 0.1, 0.3, 0.5, 0.7}
>    - **Normalization:** {None, Batch Normalization (BN), Layer Normalization (LN)}
>    - **Jumping Knowledge:** Enabled or Disabled
>    - **Residual Connections:** Enabled or Disabled
>    - **Number of Layers:** {1, 2, 3, 4, 5, 6, 7, 8, 9, 10}
>
>    The **same training environments** including:
>
>    - Dataset splits, GPU environments, optimizer, initialization, learning rate decay, batch training, maximum number of training epochs, early stopping criterion, patience, and validation frequency.
>
>    For **hyperparameters specific to each GT, which are not present in classic GNNs**, we tuned them according to the search space specified in the original GT papers:
>
>    - **GraphGPS:** the number of heads from {1, 2, 4}, MPNNs from {GCN, GraphSAGE, GAT}, positional encoding schemes from {None, LapPE, RWSE}.
>    - **NAGphormer:** the number of heads from {1, 2, 4}, the number of hops from {3, 10}.
>    - **Exphormer:** the number of heads from {1, 2, 4}, positional encoding schemes from {None, LapPE, RWSE}.
>    - **GOAT:** the number of heads from {1, 2, 4}, codebook size {1024, 2048, 4096}.
>    - **NodeFormer:** the number of heads from {1, 2, 4}, M from {30, 50}, K from {5, 10}, rb order from {1, 2}, and the temperature from {0.25}.
>    - **Polynormer:** the number of heads from {1, 2, 4, 8}.
>    - **SGFormer:** the number of heads from {1, 2, 4}, graph weight within {0.5, 0.8}.
>
>    In our original manuscript (line 172), we mentioned that "we do not perform an exhaustive search for GNNs." However, during the rebuttal period, we conducted an exhaustive search for GTs within the specified ranges. **The hyperparameter tuning range for GTs was significantly larger than that for GNNs.** Therefore, we respectfully disagree with Reviewer PM1d's assertion "while you have extensively tuned GNNs, the same has not been done with due diligence on GTs."
>
> 2. The results of GTs reported in our manuscript and the PDF in our rebuttal (whichever is better per case) represent the **best results** obtained on these datasets with the standard splits, to the best of our knowledge. **We have not encountered better results reported in existing papers or leaderboards.** This ensures that our GTs are well-tuned.

---

### Official Review · Reviewer_Ejkm · 2024-07-24
**My main concern is about correctness**

**Rating:** 7
**Confidence:** 3
**Clarity:** This paper is well-written and easy t…

**Review:**

This paper presents interesting experimental results. Generally, GTs is known to perform better than GNNs, but the finding that GNNs can outperform existing GTs with sufficient hyperparameter tuning is interesting. Additionally, analyzing how the performance changes with or without each component (e.g., dropout, normalization) can provide hints on how to design the architecture depending on the property of the dataset. However, there are concerns regarding the correctness of the two main claims mentioned earlier. Please refer to the "Correctness" section for these concerns.

**Strengths:**

- The authors confirmed the importance of hyperparameter tuning for GNNs.
- The authors demonstrated that classic GNNs can outperform recent state-of-the-art Graph Transformer (GT) models in both homophily and heterophily graphs.
- The authors performed extensive ablation studies related to the various components, such as normalization, dropout, and jumping knowledge.

**Additional Feedback:**

Please refer to the previous sections.

**Correctness:**

## Regarding the performance comparison between GTs and GNNs:

- The hyperparameter search space for GTs is not described. Did you use the hyperparameter settings from [12]? If so, it should be specified in the paper as stated.
- Consequently, based on the content of this paper, it is not possible to compare the size of hyperparameter search space used for GNNs with that of GTs. If the hyperparameter search space used for GNNs is larger than that for GTs, this comparison cannot be considered fair.
- It would have been better for the completeness of the paper if the authors had directly tuned the hyperparameters for GTs as well.
- Dropout, normalization, JK, residuals, etc., are introduced, but it would be beneficial if the main paper briefly explains whether these can be applied to GTs (or if they have already been applied or cannot be applied). Additionally, if these can be further applied, they should be included in the hyperparameter search space for GTs for a fair comparison.

## Regarding ablation analysis
- In Tables 5-7, there are many cases where the mean and standard deviation are exactly the same. It would be beneficial to verify whether this is a typographical error or if the mean and standard deviation truly match.
- If this is not a typographical error and the mean and standard deviation are indeed the same, an additional explanation is needed regarding whether the dataset contains a large number of trivially easy or difficult samples, which might lead to this phenomenon.

**Documentation:**

The paper clearly describes the hyperparameter settings, and codes are available in the supplementary material.

**Ethics:**

There are no ethics concerns.

**Limitations:**

The authors addressed the limitations in the supplementary materials. (It would be beneficial to extend our benchmarking efforts to include classic GNNs in graph-level and edge-level tasks.)
In addition, it would be more beneficial if the authors additionally focused on the node regression problem, which is also a node-level task.

**Opportunities For Improvement:**

Refer to the 'Correctness' part.

**Relation To Prior Work:**

There is no 'Related Work' section in the main paper, but the findings clearly differ from previous research results. Specifically, while most studies have reported that Graph Transformers (GTs) outperform Graph Neural Networks (GNNs), this paper demonstrates that with more sophisticated hyperparameter tuning, GNNs can actually achieve better performance.

**Summary And Contributions:**

The authors found that with proper hyperparameter tuning, contrary to the expectations of many, classic Graph Neural Networks (GNNs) can outperform recent state-of-the-art Graph Transformer (GT) models in node classification across both homophilous and heterophilous graphs. Additionally, through extensive ablation studies, they analyzed the impact of the presence or absence of various components on the performance of classic GNNs.

---

> ### Author Rebuttal · Authors · 2024-08-17
>
> **We greatly appreciate your detailed feedback and recognition of our contributions. We hope our response below effectively addresses your concerns.**
>
> **Comment 1: Correctness Regarding the Performance Comparison between GTs and GNNs**
>
> >- The hyperparameter search space for GTs is not described. Did you use the hyperparameter settings from [12]? If so, it should be specified in the paper as stated.
> >- Consequently, based on the content of this paper, it is not possible to compare the size of hyperparameter search space used for GNNs with that of GTs. If the hyperparameter search space used for GNNs is larger than that for GTs, this comparison cannot be considered fair.
> >- It would have been better for the completeness of the paper if the authors had directly tuned the hyperparameters for GTs as well.
>
> Thanks for your insightful comment! Please refer to our golabl rebuttal.
>
> >- Dropout, normalization, JK, residuals, etc., are introduced, but it would be beneficial if the main paper briefly explains whether these can be applied to GTs (or if they have already been applied or cannot be applied). Additionally, if these can be further applied, they should be included in the hyperparameter search space for GTs for a fair comparison.
>
> We apprecite your thoughful suggestion!
>
> Specifically, regarding the hyperparameters used in the main paper for GTs, to our best knowledge:
>
> - Dropout has already been applied to all GTs.
> - Normalization has already been applied to all GTs.
> - Residual connections have already been applied to all GTs.
> - JK has already been applied to some GTs like Polynormer, but it can be applied to all GTs.
>
>
> Please note that we have retrained four SOTA GTs with all the aforementioned hyperparameters, using the same search space as the classic GNNs. The updated results are provided in the one-page PDF (please refer to the global rebuttal).
>
> **Comment 2: Correctness Regarding Ablation Analysis**
>
> >- In Tables 5-7, there are many cases where the mean and standard deviation are exactly the same. It would be beneficial to verify whether this is a typographical error or if the mean and standard deviation truly match.
> >- If this is not a typographical error and the mean and standard deviation are indeed the same, an additional explanation is needed regarding whether the dataset contains a large number of trivially easy or difficult samples, which might lead to this phenomenon.
>
> We apologize for the confusion.
>
> In some experiments, the ablation results w.r.t. certain hyperparameters in Tables 5-7 of the main text mirror the GNN\* results because **those hyperparameters were not used**. We empirically found that including them may result in inferior performance. For example, in Table 5 of the main text, on the Cora dataset, the ablation results w.r.t. JK and Normalization of GCN* are exactly the same as the results of GCN*, as both hyperparameters were set to "False" (see Table 1 of Appendix A.2 in our Supplementary Material).
>
> Please note that in **Tables 1-3 of Appendix A.2**, we have provided detailed hyperparameter settings used for the three classic GNNs in Tables 2-4 of the main text. **In any case where the mean and standard deviation results remain exactly the same in Tables 5-7, it may indicate that the respective hyperparameter settings were set to "False" in Tables 1-3 of Appendix A.2.**
>
> Thank you once again for helping us improve our work!

---

> > ### Comment · Reviewer_Ejkm · 2024-08-26
> >
> > Thank you for your clarification.
> >
> > Regarding Comment 1, my concern has been resolved.
> >
> > Regarding Comment 2, I hope the authors consider a more improved presentation method for representing the result in Tables 5-7 to prevent other readers from misunderstanding this as well.

---

> > ### Author Response · Authors · 2024-08-26
> >
> > Dear Reviewer Ejkm,
> >
> > Thank you very much for taking the time to read our response and provide further feedback！
> >
> > >I hope the authors consider a more improved presentation method for representing the result in Tables 5-7 to prevent other readers from misunderstanding this as well.
> >
> > We have updated the tables to improve clarity and have included them in Tables 1-3 of the PDF attached to the new rebuttal for your review. These changes will be incorporated into the revised version.
> >
> > Since your main concern regarding the correctness of the evaluation has been resolved, would you kindly consider increasing your overall rating of our work (your initial rating was slightly negative)? If you have any remaining concerns, we would be more than happy to address them.
> >
> > We greatly appreciate your effort in helping us improve our work and your recognition of its strengths.
> >
> > Best Regards,
> >
> > Authors

---

> > > ### Author Rebuttal · Authors · 2024-08-26
> > >
> > > We attach a one-page PDF that contains the updated tables for the ablation study.

---

> > > > ### Author Response · Authors · 2024-08-30
> > > >
> > > > Dear Reviewer Ejkm,
> > > >
> > > > Thank you once again for acknowledging our rebuttal that addresses your first concern.
> > > >
> > > > As the author-reviewer discussion period will close in two days, we would greatly appreciate it if you could review our further response and rebuttal posted on August 26, where we addressed your second concern regarding the presentation of the results from our ablation study.
> > > >
> > > > We earnestly hope that you will consider updating your rating to reflect the current state of the discussion. Your support is crucial in raising awareness and promoting higher standards and more rigorous evaluations in graph machine learning.
> > > >
> > > > If you have any additional concerns, please let us know. Thank you for your time and consideration.
> > > >
> > > > Best Regards,
> > > >
> > > > Authors

---

> > > > > ### Comment · Reviewer_Ejkm · 2024-09-01
> > > > >
> > > > > Thank you for your response again, I have updated my score.

---

> > > > > > ### Author Response · Authors · 2024-09-01
> > > > > >
> > > > > > Dear Reviewer Ejkm,
> > > > > >
> > > > > > We sincerely appreciate you taking the time to update your score and informing us! Your support for promoting empirical rigor is invaluable to us and the graph machine learning community.
> > > > > >
> > > > > > We will ensure that all the suggested changes are incorporated into the revised version of our paper.
> > > > > >
> > > > > > Thank you once again for helping us improve our work.
> > > > > >
> > > > > > Best Regards,
> > > > > >
> > > > > > Authors

---

> > > > > > > ### Author Response · Authors · 2024-09-02
> > > > > > >
> > > > > > > Dear Reviewer Ejkm,
> > > > > > >
> > > > > > > We have noticed that you have further increased your score from 6 to 7, and we would like to express our sincere gratitude once again for your recognition of our contributions and efforts!
> > > > > > >
> > > > > > > Best Regards,
> > > > > > >
> > > > > > > Authors

---

### Official Review · Reviewer_xHX5 · 2024-07-30
**Well-presented result showing that GNNs are hard-to-beat**

**Rating:** 7
**Confidence:** 3
**Correctness:** The evaluations in the paper appear t…
**Clarity:** The paper is well-written and easy to…

**Review:**

This paper does a great job at showing that older architectures can often perform substantially better than new architectures on node classification problems, so long as one tunes the hyperparameters. The authors appear to tune hyperparameters for the older GNN architectures in their study, which achieve SOTA performance, but it remains unclear if the authors also tuned the graph transformer baselines that they consider. With that said, the rest of the evaluation of GNNs is exhaustive and presented in a very clear way.

**Strengths:**

- The paper is well-written and the contribution is immediately clear: the authors show that with proper hyperparameter tuning, GNNs outperform Graph Transformers, and the authors conduct an extensive ablation to identify which hyperparameters consistently contribute to improvements in performance.
- The evaluation is thorough, and involves 18 node classification datasets, with discussions of the performance across both homophilous and heterophilous graphs.
- A wide variety of methods are evaluated on these tasks.
- The list of observations and key takeaways in the experiments section are helpful for clarity and further add to the paper's readability.

**Additional Feedback:**

My main concern is about whether or not the graph transformer baselines have also been tuned. The authors should address this point or discuss it somewhere in the paper (if this is already in the paper I might have missed it).

**Documentation:**

Sufficient detail is given for each of the datasets and hyperparameter search spaces.

**Ethics:**

No ethical concerns.

**Limitations:**

Yes the authors discuss the limitations of their work, in that their hyperparameter search space was limited and that they only consider node classification tasks.

**Opportunities For Improvement:**

- It was unclear on my first read whether or not the graph transformer methods were also allocated the same tuning budget.
- The actual hyperparameter tuning pipeline and budget should be formalized as part of the benchmark itself for future methods to make an apples-to-apples comparison to GNNs on these tasks.
- It would be helpful to also include some kind of ablation of graph transformers for comparison. Do the same hyperparameters that are found to be useful for GNNs help for graph transformers?

**Relation To Prior Work:**

The prior work is adequately discussed.

**Summary And Contributions:**

This paper re-evaluates classic GNN architectures on a variety of 18 diverse node classification datasets, and finds that with tuning, classic GNNs outperform their more sophisticated counterparts such as Graph Transformers. Additionally, the authors find that GNNs achieve SOTA on the majority of the evaluated node classification tasks when tuned properly.

---

> ### Author Rebuttal · Authors · 2024-08-17
>
> **We greatly appreciate the very detailed feedback and your recognition of our contributions!**
>
> **Comment 1: Clarification on Hyperparameter Tuning Budget of GTs**
>
> > It was unclear on my first read whether or not the graph transformer methods were also allocated the same tuning budget.
> >
> > My main concern is about whether or not the graph transformer baselines have also been tuned. The authors should address this point or discuss it somewhere in the paper (if this is already in the paper I might have missed it).
>
> We appreciate your insightful comment! Please refer to our golabl rebuttal.
>
> **Comment 2: Tuning Pipeline for Benchmarking**
>
> > The actual hyperparameter tuning pipeline and budget should be formalized as part of the benchmark itself for future methods to make an apples-to-apples comparison to GNNs on these tasks.
>
> Many thanks for your thoughful suggestion! We provided a detailed report of our hyperparameter search space in lines 166-172 of the main text, with specific hyperparameter selections detailed in Tables 1-3 of Appendix A.2. Note that (as stated in line 172) we did not perform an exhaustive search for the classic GNNs; thus, conducting a comprehensive search could potentially yield better results.
>
> We believe that an automated end-to-end hyperparameter tuning pipeline is essential for benchmarking new GNN or GT models. We will endeavor to achieve this and will attempt to collaborate with the Open Graph Benchmark (OGB) team to make this pipeline available. We hope our efforts will promote more rigorous empirical evaluations in graph machine learning research.
>
> **Comment 3: Ablation Study of GTs**
>
> > It would be helpful to also include some kind of ablation of graph transformers for comparison. Do the same hyperparameters that are found to be useful for GNNs help for graph transformers?
>
> Thank you for the thoughful suggestion. Following your advice, we conducted additional ablation experiments to assess the impact of Normalization, Dropout, Residual Connections, and Jumping Knowledge—on the best-performing GT, Polynormer. The resuls below demonstrate that the hyperparameters that are found to be useful for GNNs, like Dropout and Residual Connections, are also beneficial for GTs.
>
> |                          | Computer↑ (5 layers) | CS↑ (5 layers) | Amazon-Ratings↑ (10 layers) | Minesweeper↑ (10 layers) | ogbn-arxiv↑ (7 layers) |
> | ------------------------ | -------------------- | -------------- | --------------------------- | ------------------------ | ---------------------- |
> | Graph Type               | Homophily            | Homophily      | Heterophily                 | Heterophily              | Large graphs           |
> | **Polynormer***          | 93.78 ± 0.10         | 95.42 ± 0.19   | 54.96 ± 0.22                | 97.49 ± 0.48             | 73.40 ± 0.22           |
> | (-) Normalization        | 93.66 ± 0.08         | 95.42 ± 0.19   | 54.96 ± 0.22                | 97.49 ± 0.48             | 71.81 ± 0.46           |
> | (-) Dropout              | 91.37 ± 0.33         | 95.25 ± 0.12   | 54.40 ± 0.35                | 93.93 ± 0.22             | 72.44 ± 0.35           |
> | (-) Residual Connections | 91.02 ± 3.34         | 94.12 ± 0.13   | 48.75 ± 0.21                | 90.63 ± 0.51             | 72.84 ± 0.18           |
> | (-) Jumping Knowledge    | 93.29 ± 0.54         | 94.86 ± 0.56   | 54.52 ± 0.31                | 96.80 ± 0.83             | 73.08 ± 0.29           |
>
>
> We will incorporate the results in the revised version. Thank you once again for helping us improve our work!

---

### Author Rebuttal · Authors · 2024-08-17

**We express our gratitude to all reviewers for their invaluable time, effort, and the comprehensive, constructive feedback!** We are happy to provide additional information if needed.

We have noted that the reviewers' concerns generally center around the hyperparameter tuning of Graph Transformers (GTs). To address this concern, we've uploaded a one-page PDF containing new experimental results and wish to provide further clarification on this matter.

-----------------------------------------

**G1 Retraining the GTs using the same hyperparameter tuning and training environments as the classic GNNs** (xHX5 C1, Ejkm C1, PM1d C1, madU Q2)

We greatly apprecite the reviewers' thoughtful suggestion!

It is important to note that we reported the results for all GTs from their original papers or established leaderboards whenever possible (as discussed in lines 214-218), as we believed those results had been optimized to their best performance. Thanks to the reviewers' comments, we recognize the importance of benchmarking them within the same hyperparameter search space and under consistent enviorments. We are motivated and pleased to provide updated results.

Due to the time constraints of the rebuttal period, we retrained 4 state-of-the-art GTs (GraphGPS, NodeFormer, SGFormer, Polynormer) using **the same hyperparameter tuning search space and training environments** as the classic GNNs (as detailed in lines 166-172). Additionally, for the hyperparameters specific to each GT (which are not present in the classic GNNs), we tuned them according to the search space specified in the original GT paper. We provide partial results in the table below and the **complete results available in Tables 1-3 in the uploaded PDF**. (edit: we identified a typo in the results for SGFormer and have updated it accordingly.)

These results show **minor variations compared to the previously reported results**. The only excecptions are NodeFormer on ogbn-arxiv and Roman-Empire, and SGFormer on WikiCS, ogbn-products and pokec, where their performance has significantly improved, although they still remain considerably worse than that of GNNs. The updated results **reaffirm our conclusion** that classic GNNs can achieve state-of-the-art performance in node classification tasks.

We will continue to retrain the other baseline GTs  and provide updated results in the coming days during the discussion period. These results will be included in the revised version of our manuscript.

|                 | Cora↑            | CiteSeer↑        | PubMed↑          | Computer↑        | Photo↑           | CS↑              | Physics↑         | WikiCS↑          | ogbn-proteins↑   | ogbn-arxiv↑      | ogbn-products↑   | pokec↑           |
| --------------- | ---------------- | ---------------- | ---------------- | ---------------- | ---------------- | ---------------- | ---------------- | ---------------- | ---------------- | ---------------- | ---------------- | ---------------- |
| **GraphGPS***   | 83.87 ± 0.96     | 72.73 ± 1.23     | 79.94 ± 0.26     | 91.79 ± 0.63     | 94.89 ± 0.14     | 94.04 ± 0.21     | 96.71 ± 0.15     | 78.66 ± 0.49     | 77.15 ± 0.64     | 71.23 ± 0.59     | OOM              | OOM              |
| **NodeFormer*** | 82.73 ± 0.75     | 72.37 ± 1.20     | 79.59 ± 0.92     | 87.29 ± 0.58     | 93.43 ± 0.56     | 95.69 ± 0.27     | 96.48 ± 0.34     | 75.13 ± 0.93     | 77.86 ± 0.84     | 67.78 ± 0.28     | 73.96 ± 0.30     | 71.00 ± 1.30     |
| **SGFormer***   | 84.82 ± 0.85     | 72.72 ± 1.15     | 80.60 ± 0.49     | 92.42 ± 0.66     | 95.58 ± 0.36     | 95.71 ± 0.24     | 96.75 ± 0.26     | 80.05 ± 0.46     | 79.92 ± 0.48     | 72.76 ± 0.33     | 81.54 ± 0.43     | 82.44 ± 0.76     |
| **Polynormer*** | 83.43 ± 0.89     | 72.19 ± 0.83     | 79.35 ± 0.73     | 93.78 ± 0.10     | 96.57 ± 0.23     | 95.42 ± 0.19     | 97.18 ± 0.11     | 80.26 ± 0.92     | 79.53 ± 0.67     | 73.40 ± 0.22     | 83.82 ± 0.11     | 86.06 ± 0.25     |
| **GCN***        | **85.08 ± 0.52** | **72.98 ± 0.84** | **81.32 ± 0.72** | 93.80 ± 0.29     | 96.51 ± 0.20     | 95.80 ± 0.28     | **97.43 ± 0.05** | 80.27 ± 0.71     | 77.29 ± 0.46     | **73.60 ± 0.18** | 82.33 ± 0.19     | **86.33 ± 0.17** |
| **GraphSAGE***  | 84.18 ± 0.81     | 71.93 ± 0.85     | 79.41 ± 0.53     | 93.59 ± 0.22     | 96.41 ± 0.17     | **96.12 ± 0.24** | 97.21 ± 0.05     | 80.51 ± 0.48     | 82.21 ± 0.32     | 72.95 ± 0.31     | **83.89 ± 0.36** | 85.97 ± 0.21     |
| **GAT***        | 84.64 ± 1.27     | 72.10 ± 1.10     | 79.70 ± 0.70     | **93.93 ± 0.16** | **96.67 ± 0.13** | 96.08 ± 0.10     | 97.30 ± 0.06     | **80.75 ± 0.74** | **85.01 ± 0.46** | 73.30 ± 0.16     | 80.99 ± 0.16     | 86.19 ± 0.23     |

-----------------------------------------

Once again, we sincerely thank the reviewer for their invaluable feedback in helping us improve our work!

---

### Author Rebuttal · Authors · 2024-08-20

Dear Reviewers,

We would like to provide an update that we have retrained all baseline GTs including NAGphormer, Exphormer, and GOAT using **the same hyperparameter tuning search space and training environments** as those used for the classic GNNs. Please note that "same training environments" include using **identical initialization seeds, train/val/test splits, and GPU environments**. Partial results are provided in the table below, with the **complete results available in Tables 1-3 in the PDF attached to this rebuttal**.

Similar observations can be drawn from the additional results. Notably, only the performance of GOAT on the ogbn-proteins and pokec datasets exhibits significant improvements over the previously reported results, although these improvements still do not reach the levels achieved by the GNNs like GAT or GraphSAGE. The remaining results display only **minor variations**.

Please note that **we have updated all the results of the GTs** in our paper, adhering to the reviewers' suggestions and ensuring that the **GTs are tuned to their optimal performance**, thereby **guaranteeing fair and equitable comparisons**. Based on these results, we believe it is reasonable to conclude that classic GNNs can achieve performance that is competitive with, or even superior to, that of the GTs, on node classification tasks.

|                 | Cora↑            | CiteSeer↑        | PubMed↑          | Computer↑        | Photo↑           | CS↑              | Physics↑         | WikiCS↑          | ogbn-proteins↑   | ogbn-arxiv↑      | ogbn-products↑   | pokec↑           |
| --------------- | ---------------- | ---------------- | ---------------- | ---------------- | ---------------- | ---------------- | ---------------- | ---------------- | ---------------- | ---------------- | ---------------- | ---------------- |
| **GraphGPS***   | 83.87 ± 0.96     | 72.73 ± 1.23     | 79.94 ± 0.26     | 91.79 ± 0.63     | 94.89 ± 0.14     | 94.04 ± 0.21     | 96.71 ± 0.15     | 78.66 ± 0.49     | 77.15 ± 0.64     | 71.23 ± 0.59     | OOM              | OOM              |
| **NodeFormer*** | 82.73 ± 0.75     | 72.37 ± 1.20     | 79.59 ± 0.92     | 87.29 ± 0.58     | 93.43 ± 0.56     | 95.69 ± 0.27     | 96.48 ± 0.34     | 75.13 ± 0.93     | 77.86 ± 0.84     | 67.78 ± 0.28     | 73.96 ± 0.30     | 71.00 ± 1.30     |
| **SGFormer***   | 84.82 ± 0.85     | 72.72 ± 1.15     | 80.60 ± 0.49     | 92.42 ± 0.66     | 95.58 ± 0.36     | 95.71 ± 0.24     | 96.75 ± 0.26     | 80.05 ± 0.46     | 79.92 ± 0.48     | 72.76 ± 0.33     | 81.54 ± 0.43     | 82.44 ± 0.76     |
| **Polynormer*** | 83.43 ± 0.89     | 72.19 ± 0.83     | 79.35 ± 0.73     | 93.78 ± 0.10     | 96.57 ± 0.23     | 95.42 ± 0.19     | 97.18 ± 0.11     | 80.26 ± 0.92     | 79.53 ± 0.67     | 73.40 ± 0.22     | 83.82 ± 0.11     | 86.06 ± 0.25     |
| **NAGphormer*** | 80.92 ± 1.17     | 70.59 ± 0.89     | 80.14 ± 1.06     | 91.69 ± 0.30     | 96.14 ± 0.16     | 95.85 ± 0.16     | 97.35 ± 0.12     | 77.92 ± 0.93     | 72.17 ± 0.45     | 70.88 ± 0.24     | 74.63 ± 0.29     | 75.92 ± 0.68     |
| **Exphormer***  | 83.29 ± 1.36     | 71.85 ± 1.11     | 79.67 ± 0.73     | 91.80 ± 0.35     | 95.69 ± 0.39     | 95.92 ± 0.25     | 97.06 ± 0.13     | 79.38 ± 0.62     | 77.62 ± 0.33     | 72.32 ± 0.36     | OOM              | OOM              |
| **GOAT***       | 83.26 ± 1.24     | 72.21 ± 1.29     | 80.06 ± 0.67     | 92.29 ± 0.37     | 94.33 ± 0.21     | 93.81 ± 0.19     | 96.47 ± 0.16     | 77.96 ± 0.63     | 79.31 ± 0.42     | 72.76 ± 0.29     | 82.27 ± 0.56     | 72.64 ± 0.67     |
| **GCN***        | **85.08 ± 0.52** | **72.98 ± 0.84** | **81.32 ± 0.72** | 93.80 ± 0.29     | 96.51 ± 0.20     | 95.80 ± 0.28     | **97.43 ± 0.05** | 80.27 ± 0.71     | 77.29 ± 0.46     | **73.60 ± 0.18** | 82.33 ± 0.19     | **86.33 ± 0.17** |
| **GraphSAGE***  | 84.18 ± 0.81     | 71.93 ± 0.85     | 79.41 ± 0.53     | 93.59 ± 0.22     | 96.41 ± 0.17     | **96.12 ± 0.24** | 97.21 ± 0.05     | 80.51 ± 0.48     | 82.21 ± 0.32     | 72.95 ± 0.31     | **83.89 ± 0.36** | 85.97 ± 0.21     |
| **GAT***        | 84.64 ± 1.27     | 72.10 ± 1.10     | 79.70 ± 0.70     | **93.93 ± 0.16** | **96.67 ± 0.13** | 96.08 ± 0.10     | 97.30 ± 0.06     | **80.75 ± 0.74** | **85.01 ± 0.46** | 73.30 ± 0.16     | 80.99 ± 0.16     | 86.19 ± 0.23     |


Thank you for taking the time to review our response! We would be more than happy to answer any further questions you may have.

Best Regards,

Authors

---

### Decision · Program_Chairs · 2024-09-26

**Decision:**

Accept (Poster)

**Comment:**

This paper conducts a thorough empirical analysis to reevaluate the performance of three classic GNN models (GCN, GAT, and GraphSAGE) against GTs. The paper finds that GNNs achieve SOTA on the majority of the evaluated node classification tasks when tuned properly. The paper is well-written. The evaluation is through, which involves 18 node classification datasets. The findings are interesting and important. The authors clarified some unclear points and added extra experiments, which addressed most of the reviewers’ concerns. The authors are encouraged to include these explanations and experiments in the camera ready version.